# NON-ASYMPTOTIC ANALYSIS OF (STICKY) TRACK-AND-STOP

**Riccardo Poiani, Martino Bernasconi & Andrea Celli**
Bocconi University
Milan, Italy
`{riccardo.poiani,martino.bernasconi,andrea.celli2}@unibocconi.it`

## ABSTRACT

In pure exploration problems, a statistician sequentially collects information to answer a question about some stochastic and unknown environment. The probability of returning a wrong answer should not exceed a maximum risk parameter $\delta$ and good algorithms make as few queries to the environment as possible. The Track-and-Stop algorithm is a pioneering method to solve these problems. Specifically, it is well-known that it enjoys asymptotic optimality sample complexity guarantees for $\delta \to 0$ whenever the map from the environment to its correct answers is single-valued (*e.g.*, best-arm identification with a unique optimal arm). The Sticky Track-and-Stop algorithm extends these results to settings where, for each environment, there might exist multiple correct answers (*e.g.*, $\epsilon$-optimal arm identification). Although both methods are optimal in the asymptotic regime, their non-asymptotic guarantees remain unknown. In this work, we fill this gap and provide non-asymptotic guarantees for both algorithms.

## 1 INTRODUCTION

In pure exploration problems, a statistician interacts with a set of $K \in \mathbb{N}$ probability distributions denoted by $\boldsymbol{\varphi} = \{\varphi_i\}_{i \in [K]}$, commonly referred to as arms. Their unknown means are denoted by $\boldsymbol{\mu} = \{\mu_i\}_{i \in [K]}$, and $\boldsymbol{\mu}$ belongs to some set $\mathcal{M} \subseteq \mathbb{R}^K$ which encodes some possibly known structure among the different arms, *e.g.*, Lipschitzianity (Wang et al., 2021) or unimodality (Poiani et al., 2024). During each step $t \in \mathbb{N}$, the statistician chooses an arm $A_t$, and observes a sample $X_t \sim \varphi_{A_t}$ from the corresponding reward distribution. Given a maximum risk parameter $\delta \in (0, 1)$, the statistician aims to answer a question about the unknown means $\boldsymbol{\mu}$ while using as few samples as possible. Specifically, there is a known answer space $\mathcal{I}$ and a (set-valued) answer function $i^\star(\boldsymbol{\mu})$ that maps each bandit $\boldsymbol{\mu}$ to a subset of correct answers $i^\star(\boldsymbol{\mu})$ within $\mathcal{I}$. The probability of returning an answer that does not belong to $i^\star(\boldsymbol{\mu})$ should not exceed $\delta$.

The most studied pure exploration problem is Best-Arm Identification (BAI, Garivier & Kaufmann, 2016), where the answer space is $\{1, \ldots, K\}$ and the goal is to return the unique index of the arm with the highest mean, *i.e.*, $i^\star(\boldsymbol{\mu}) = \operatorname{argmax}_{k \in [K]} \mu_k$. The pioneering work by Garivier & Kaufmann (2016) developed a tight lower bound for the unstructured BAI problem and proposed the Track-and-Stop (TAS) algorithm to solve it. Remarkably, the expected number of samples required by TAS to identify $i^\star(\boldsymbol{\mu})$ with high probability exactly matches the lower bound as $\delta$ approaches 0. In this sense, TAS is asymptotically optimal for $\delta \to 0$. These results have been shown to hold even outside the BAI setting. Indeed, they extend to the more general structured partition identification problem (Kaufmann & Koolen, 2021). Here, $\mathcal{M}$ is partitioned into a finite number $|\mathcal{I}|$ of disjoint subsets, *i.e.*, $\mathcal{M} = \bigcup_{i \in \mathcal{I}} \mathcal{M}_i$, and the statistician aims to find $i$ such that $\boldsymbol{\mu} \in \mathcal{M}_i$. In this sense, TAS turned out to be a powerful tool that can be used to solve a wide variety of problems (*i.e.*, all the problems where $i^\star(\boldsymbol{\mu})$ is unique) while enjoying asymptotic optimality. If there exists multiple correct answers for a certain bandit $\boldsymbol{\mu}$, *i.e.*, $i^\star(\boldsymbol{\mu})$ is multi-valued, TAS fails to achieve optimality (Degenne & Koolen, 2019). This is the case, for instance, of the $\epsilon$-best arm identification problem, where multiple arms $j$ might satisfy $\mu_j \geq \operatorname{argmax}_{k \in [k]} \mu_k - \epsilon$. To solve this issue, Degenne & Koolen (2019) proposed a modification of the TAS algorithm called Sticky Track-and-Stop (S-

TAS). S-TAS enjoys asymptotic optimality for any pure exploration problem with multiple correct answers.

Due to their generality and strong theoretical guarantees, both TAS and S-TAS have become fundamental algorithms. However, as noted in several works, the analysis of TAS is asymptotic in nature and does not offer insights into its non-asymptotic behavior (Barrier et al., 2022; Barrier, 2023; Jourdan & Degenne, 2023; Poiani et al., 2024; Russo et al., 2025). Barrier (2023) suggest that this difficulty arises from instability in the sampling rule when data are scarce: the sampling rule employed by TAS can vary significantly early on, when estimated means have not yet concentrated around their expectations. The problem is even more pronounced for S-TAS. Intuitively, this is due to the fact that the algorithm is more complex, and that it samples the next arm in a TAS fashion.

In this work, we address the following question:

*Can we characterize the non-asymptotic guarantees of* TAS *and* S-TAS*?*

We answer this question by providing the first non-asymptotic bounds for both Track-and-Stop and Sticky Track-and-Stop, shedding light on their behavior in the finite-confidence setting. Our results hold for arbitrarily structured problems and correct answer correspondences.

Importantly, from an empirical side, the performance of TAS has been benchmarked several times (*e.g.*, Degenne et al., 2019; Wang et al., 2021; Jourdan et al., 2022; Barrier et al., 2022). The results have consistently shown that TAS obtains highly competitive sample complexity even in the moderate regime of $\delta$. Thus, the theoretical findings that we present here *complement* these results, showing that TAS also enjoys finite-confidence guarantees. Furthermore, since S-TAS is, to the best of the authors' knowledge, the only algorithm in the literature that can solve arbitrary multiple answer problems, our work also provides the *first finite-confidence analysis for this general class of problems*.

## 1.1 RELATED WORK

**Single-Answer Problems and TAS** Since the work by Garivier & Kaufmann (2016), which studied the unstructured bandit problem, several works have extended TAS to several structured problems (*e.g.*, Moulos, 2019; Juneja & Krishnasamy, 2019; Kocák & Garivier, 2020; Kaufmann & Koolen, 2021; Poiani et al., 2024; Kanarios et al., 2024). In these works, the optimality analysis of TAS remained asymptotic, as they build upon the approach of Garivier & Kaufmann (2016). Beyond TAS, several other algorithms have been proposed in the literature that achieve asymptotic optimality for best-arm identification and/or single-answer problems (Degenne et al., 2019; Ménard, 2019; Wang et al., 2021; Barrier et al., 2022; Jourdan et al., 2022; Jourdan & Degenne, 2023). Moreover, some of these works also establish non-asymptotic upper bounds on the expected number of samples required for their algorithm to stop (Degenne et al., 2019; Barrier et al., 2022; Jourdan & Degenne, 2023; Wang et al., 2021). Among these studies, the works most closely related to ours are Degenne et al. (2019) and Barrier et al. (2022), which propose two distinct approaches that solve the aforementioned instability issue of TAS. Precisely, Degenne et al. (2019) propose an optimistic version of TAS that incorporates confidence intervals within the sampling rule. This comes at the cost of solving a significantly more challenging optimization problem in deciding the next arm to query. In particular, while TAS bases its decision rule on a maximization problem of a convex function over the simplex, its optimistic version requires solving a max-max-min problem which, in general, does not admit efficient oracles. Barrier et al. (2022), instead, stabilizes the sampling rule by "skewing" its behavior toward uniform exploration when the amount of data collected is scarce. This modification, however, can lead to a decrease in the empirical performance w.r.t. the original version. Indeed, Barrier et al. (2022) show that the original version of TAS achieves better results in all instances in which the algorithms were compared. Hence, given these remarks, it remains important to understand whether finite confidence guarantees can be obtained for the original version of TAS. In this sense, we note that both the analyses of Degenne et al. (2019) and Barrier et al. (2022) rely on specific properties of the algorithms they introduce, which makes them different from the original framework of Garivier & Kaufmann (2016). To the best of our knowledge, our work is the first to provide non-asymptotic guarantees for the original TAS algorithm without requiring any substantial modifications to it.

**Multiple-Answer Problems and S-TAS**   When there are multiple correct answers, S-TAS (Degenne & Koolen, 2019) offers a solution to pure exploration problems while enjoying asymptotic optimality. However, to the best of our knowledge, there is no variant of S-TAS that achieves finite-confidence guarantees in arbitrary multiple-answer problems. More broadly, we are not aware of any algorithm providing finite-confidence guarantees for general multiple-answer problems. Prior work has largely focused on *specific subclasses* of pure exploration problems. Among these, the most studied one is $\epsilon$-best arm identification (e.g., Even-Dar et al., 2002; Kalyanakrishnan et al., 2012; Karnin et al., 2013; Kocák & Garivier, 2021; Jourdan et al., 2023; Jin et al., 2024). Nonetheless, none of these studies can be applied to the general pure exploration setting of Degenne & Koolen (2019).

## 2   BACKGROUND

We focus on bandit problems $\boldsymbol{\varphi} = \{\varphi_k\}_{k \in [K]}$ with $K \in \mathbb{N}$ arms, where $\varphi_k$ is a probability distribution with mean $\mu_k$. We denote by $\boldsymbol{\mu} = \{\mu_k\}_{k \in [K]}$ the vector of the means of the distributions. As usual in the literature (see *e.g.*, Garivier & Kaufmann (2016); Degenne et al. (2019)), we focus on distributions that belong to a canonical exponential family[1] It is well known that such distributions are fully characterized by their means. For convenience, we will refer to a bandit model $\boldsymbol{\varphi}$ directly by its vector of means $\boldsymbol{\mu}$. We denote by $\Theta \subseteq \mathbb{R}$ an open set that defines the possible means of the distributions. We consider the general case where $\boldsymbol{\mu} \in \mathcal{M} \subseteq \Theta^K$. This allows to include in our analysis also structured settings such as Lipschitz (Wang et al., 2021) or unimodal bandits (Poiani et al., 2024). Indeed, since $\mathcal{M}$ is any subset of $\Theta^K$, it can directly encode the constraints imposed by these structures.[2] Moreover, we assume a finite answer space $\mathcal{I}$, along with access to a set-valued function $i^\star : \mathcal{M} \rightrightarrows \mathcal{I}$, which maps each model $\boldsymbol{\mu} \in \mathcal{M}$ to the set of all the answers that are correct for the bandit instance $\boldsymbol{\mu}$.

At each step $t \in \mathbb{N}$, the learner chooses an action $A_t \in [K]$ and observes a sample $X_t \sim \varphi_{A_t}$. Let $\mathcal{F}_t = \sigma(A_1, X_1, \dots A_t, X_t)$ be the $\sigma$-field generated by the interactions with the bandit model up to time $t$. Then, a pure exploration algorithm receives as input a confidence level $\delta \in (0, 1)$, and implements the following procedures: (i) a $\mathcal{F}_{t-1}$-measurable *sampling rule* which selects the action $A_t \in [K]$ based on the past observations, (ii) a *stopping rule* $\tau_\delta$ which is a stopping time w.r.t. $(\mathcal{F}_t)_{t \in \mathbb{N}}$ and controls the end of the data acquisition phase, and (iii) a $\mathcal{F}_{\tau_\delta}$-measurable *recommendation rule* $\hat{i}_{\tau_\delta} \in \mathcal{I}$ that denotes the guess of the statistician for a correct answer for $\boldsymbol{\mu}$. A pure exploration algorithm is $\delta$-*correct* on $\mathcal{M}$ if it satisfies $\mathbb{P}_{\boldsymbol{\mu}}(\hat{i}_{\tau_\delta} \notin i^\star(\boldsymbol{\mu})) \le \delta$ for all $\boldsymbol{\mu} \in \mathcal{M}$. The goal is building algorithms which are $\delta$-correct and that minimize the expected stopping time, *i.e.*, $\mathbb{E}_{\boldsymbol{\mu}}[\tau_\delta] = \sum_{k \in [K]} \mathbb{E}_{\boldsymbol{\mu}}[N_k(\tau_\delta)]$, where $N_k(t)$ is the number of samples collected for arm $k \in [K]$ up to time $t$. In the following, we denote by $\boldsymbol{N}(t)$ the vector $(N_1(t), \dots, N_K(t))$.

**Additional Notation**   For a given set $\mathcal{X}$, we denote by $\mathrm{cl}(\mathcal{X})$ its closure. Furthermore, for all $i \in \mathcal{I}$, we denote by $\neg i = \{\boldsymbol{\lambda} \in \mathcal{M} : i \notin i^\star(\boldsymbol{\lambda})\}$. In words, $\neg i$ represents the set of bandit models for which $i$ is not a correct answer. Without loss of generality, we assume that for all $\boldsymbol{\mu} \in \mathcal{M}$, there exists $i \in i^\star(\boldsymbol{\mu})$ such that $\boldsymbol{\mu} \notin \mathrm{cl}(\neg i)$.[3] Furthermore, for distributions with means $p$ and $q$, we write $d(p, q)$ to denote their KL divergence. Moreover, for a distribution with mean $p$, we denote by $\nu_p$ the corresponding natural parameter within the exponential family. For $n \in \mathbb{N}$, $\Delta_n$ denotes the $n$-dimensional simplex. Finally, consider two topological spaces $\mathcal{X}$ and $\mathcal{Y}$, and consider a set-valued function $F : \mathcal{X} \rightrightarrows \mathcal{Y}$ that maps each $x \in \mathcal{X}$ to $F(x) \subseteq \mathcal{Y}$. We say that $F$ is upper hemicontinuous, if for all $x \in \mathcal{X}$ and any open set $\mathcal{V} \subseteq \mathcal{Y}$ such that $F(x) \subseteq \mathcal{V}$, there exists a neighbourhood $\mathcal{U}$ of $x$ such that, for all $x' \in \mathcal{U}$, $F(x')$ is a subset of $\mathcal{Y}$ (Aubin, 1999).

**Lower Bound for Single-Answer Problems**   Let us focus on the case where $i^\star(\boldsymbol{\mu})$ is unique for all $\boldsymbol{\mu} \in \mathcal{M}$. Lower bounds for these problems can be derived following the arguments presented

---

[1]These include Gaussian with known variance and Bernoulli distributions. See Cappé et al. (2013).

[2]For completeness, we show in Appendix A how to encode these structures through $\mathcal{M}$.

[3]It is easy to verify from the lower bounds that, whenever this assumption is not satisfied, one obtains infinite sample complexity. This requirement is usually implicitly satisfied in the literature, *e.g.*, $\mathrm{argmax}_{k \in [K]} \mu_k$ is unique over $\mathcal{M}$ in best-arm identification problems (Degenne & Koolen, 2019), and the different sets $\mathcal{M}_j$ are *open* and disjoint in the partition identification problem (Wang et al., 2021).

in Garivier & Kaufmann (2016). Specifically, for any $\delta$-correct algorithm, it holds that $\mathbb{E}_{\boldsymbol{\mu}}[\tau_\delta] \geq T^\star(\boldsymbol{\mu}) \log(1/(2.4\delta))$ (see Appendix B for a formal statement), where:

$$T^\star(\boldsymbol{\mu})^{-1} = \sup_{\boldsymbol{\omega} \in \Delta_K} \inf_{\boldsymbol{\lambda} \in \neg i^\star(\boldsymbol{\mu})} \sum_{k \in [K]} \omega_k d(\mu_k, \lambda_k) \tag{1}$$

$$= \sup_{\boldsymbol{\omega} \in \Delta_K} \max_{i \in \mathcal{I}} \inf_{\boldsymbol{\lambda} \in \neg i} \sum_{k \in [K]} \omega_k d(\mu_k, \lambda_k). \tag{2}$$

$T^\star(\boldsymbol{\mu})^{-1}$ can be interpreted as a max-min game where the max player plays a sampling strategy $\boldsymbol{\omega}$ to quickly identify the correct answer $i^\star(\boldsymbol{\mu})$, and the min-player chooses a confounding instance $\boldsymbol{\lambda} \in \mathcal{M}$ where the correct answer changes (Degenne et al., 2019). The convex set of weights that attains the argmax in $T^\star(\boldsymbol{\mu})^{-1}$ are denoted by $\boldsymbol{\omega}^\star(\boldsymbol{\mu})$ and takes the name of *oracle weights*. Here, convexity simply follows from the fact $T^\star(\boldsymbol{\mu})^{-1}$ is a supremum over functions that are linear in $\boldsymbol{\omega}$. We note that we provided two expressions for $T^\star(\boldsymbol{\mu})^{-1}$. Equation (1) is the one that most frequently appears in the literature (*e.g.*, Garivier & Kaufmann (2016)). Equation (2) is a rewriting that allows to generalize the expression of $T^\star(\boldsymbol{\mu})^{-1}$ to bandit models $\boldsymbol{\mu}$'s that fall outside $\mathcal{M}$, as $i^\star(\boldsymbol{\mu})$ is formally defined only for $\boldsymbol{\mu} \in \mathcal{M}$. This is important from an algorithmic perspective as it allows us to generalize the definition of oracle weights to models that are not in $\mathcal{M}$.[4] Furthermore, as we shall see, it will also play an important role in our analysis. Finally, since the problem is single-answer, then for all $\boldsymbol{\mu} \in \mathcal{M}$ the argmax over the different answers is attained only at $i = i^\star(\boldsymbol{\mu})$.

**Lower Bound for Multiple-Answer Problems** Lower bounds for multiple-answer problems were established by Degenne & Koolen (2019). Specifically, the authors shows that, for any $\delta$-correct algorithm and any $\boldsymbol{\mu} \in \mathcal{M}$, it holds that $\liminf_{\delta \to 0} \frac{\mathbb{E}_{\boldsymbol{\mu}}[\tau_\delta]}{\log(1/\delta)} \geq T^\star(\boldsymbol{\mu})$, where $T^\star(\boldsymbol{\mu})^{-1}$ is given by (formal statement in Appendix B):

$$T^\star(\boldsymbol{\mu})^{-1} = \sup_{\boldsymbol{\omega} \in \Delta_K} \max_{i \in i^\star(\boldsymbol{\mu})} \inf_{\boldsymbol{\lambda} \in \neg i} \sum_{k \in [K]} \omega_k d(\mu_k, \lambda_k) \tag{3}$$

$$= \sup_{\boldsymbol{\omega} \in \Delta_K} \max_{i \in \mathcal{I}} \inf_{\boldsymbol{\lambda} \in \neg i} \sum_{k \in [K]} \omega_k d(\mu_k, \lambda_k). \tag{4}$$

We have introduced two expressions for $T^\star(\boldsymbol{\mu})^{-1}$: one that only applies to models within $\mathcal{M}$ (Equation (3)), and another that extends the definition to models outside $\mathcal{M}$ (Equation (4)). While these results closely resemble those of single-answer problems, a few differences need to be highlighted. First, this lower bound only holds in the asymptotic regime of $\delta \to 0$. Second, the argmax in $T^\star(\boldsymbol{\mu})^{-1}$ over the different answers can be attained at multiple points. Specifically, let $i_F(\boldsymbol{\mu})$ be the set of answers that attain the argmax, *i.e.*, $i_F(\boldsymbol{\mu}) = \mathrm{argmax}_{i \in \mathcal{I}} \sup_{\boldsymbol{\omega} \in \Delta_K} \inf_{\boldsymbol{\lambda} \in \neg i} \sum_{k \in [K]} \omega_k d(\mu_k, \lambda_k)$. Then, while for single answer problems $|i_F(\boldsymbol{\mu})| = |i^\star(\boldsymbol{\mu})| = 1$ for all $\boldsymbol{\mu} \in \mathcal{M}$, in multiple answer problems it can happen that $|i_F(\boldsymbol{\mu})| > 1$. Since it plays a crucial role in our results, we emphasize that the correspondence $\boldsymbol{\mu} \mapsto i_F(\boldsymbol{\mu})$ is upper hemicontinuous (Theorem 4 of Degenne & Koolen (2019)). Finally, we mention that the oracle weights $\boldsymbol{\omega}^\star(\boldsymbol{\mu})$ are no longer a convex set when $|i_F(\boldsymbol{\mu})| > 1$. Instead, we have that $\boldsymbol{\omega}^\star(\boldsymbol{\mu}) = \bigcup_{i \in i_F(\boldsymbol{\mu})} \boldsymbol{\omega}^\star(\boldsymbol{\mu}, \neg i)$, where each element $\boldsymbol{\omega}^\star(\boldsymbol{\mu}, \neg i) := \mathrm{argmax}_{\boldsymbol{\omega} \in \Delta_K} \inf_{\boldsymbol{\lambda} \in \neg i} \sum_{k \in [K]} \omega_k d(\mu_k, \lambda_k)$ is a convex set.

**Track-and-Stop** Track-and-Stop (TAS, Garivier & Kaufmann, 2016) works as follows. After a first phase where each arm $k \in [K]$ is pulled once, TAS computes, at each round $t$, the empirical oracle weights $\boldsymbol{\omega}(t) \in \boldsymbol{\omega}^\star(\hat{\boldsymbol{\mu}}(t))$, where $\hat{\mu}_k(t) = N_k(t)^{-1} \sum_{s=1}^t \mathbf{1}\{A_s = k\} X_s$ denotes the empirical estimate of $\mu_k$ at time $t$. Then, TAS applies a tracking procedure on $\{\boldsymbol{\omega}(t)\}_t$ to select the next action. Specifically, the *C-Tracking* procedure projects each $\boldsymbol{\omega}(t)$ onto $\Delta_K^{\epsilon_s} = \Delta_K \cap [\epsilon_s, 1]^K$ according to the $\ell_\infty$ norm. This projection takes the name of *forced exploration*, as it ensures that $N_k(t) \gtrsim \sqrt{t}$ for all $k \in [K]$ for $\epsilon_t \approx t^{-1/2}$. The next action is selected as $A_{t+1} \in \mathrm{argmax}_{k \in [K]} \sum_{s=K}^t \tilde{\omega}_k(s) - N_k(t)$, where each $\tilde{\boldsymbol{\omega}}(s)$ denotes the projection of $\boldsymbol{\omega}(s)$. Regarding the stopping and recommendation rules, TAS halts as soon as $\max_{i \in \mathcal{I}} \inf_{\boldsymbol{\lambda} \in \neg i} \sum_{k \in [K]} N_k(t) d(\hat{\mu}_k(t), \lambda_k) \geq \beta_{t,\delta}$, and recommends an index that attains the argmax in the stopping rule. By calibrating the threshold $\beta_{t,\delta}$ (typically,

---

[4]Indeed, we observe that an empirical estimate of $\boldsymbol{\mu}$ might not belong to $\mathcal{M}$.

$\beta_{t,\delta} \approx \log(1/\delta) + K \log(t)$; see *e.g.*, Kaufmann & Koolen (2021) for a complete expression of $\beta_{t,\delta}$) one can prove that those stopping and recommendation rules yield $\delta$-correctness (both for single and multiple-answer problems) when paired with *any* sampling rule.[5] TAS enjoys asymptotic optimality guarantees whenever $|i_F(\boldsymbol{\mu})| = 1$, *i.e.*, $\limsup_{\delta \to 0} \frac{\mathbb{E}_{\boldsymbol{\mu}}[\tau_\delta]}{\log(1/\delta)} \leq T^\star(\boldsymbol{\mu})$. However, this does not hold for $|i_F(\boldsymbol{\mu})| > 1$ (Degenne & Koolen, 2019). The reason is that its sampling rule ensures that the empirical pull strategy $\boldsymbol{N}(t)/t$ converges (on a good event) to the convex hull of the oracle weights, *i.e.*, $\inf_{\boldsymbol{\omega} \in \text{conv}(\boldsymbol{\omega}^\star(\boldsymbol{\mu}))} \|\boldsymbol{N}(t)/t - \boldsymbol{\omega}\| \to 0$.[6] When $|i_F(\boldsymbol{\mu})| = 1$, this convex hull coincides with $\boldsymbol{\omega}^\star(\boldsymbol{\mu})$, and this leads to optimality. However, this is not generally true in the context of multiple-answer problems.

**Sticky Track-and-Stop** To solve this issue, Degenne & Koolen (2019) proposed the Sticky Track-and-Stop (S-TAS) algorithm. The stopping and recommendation rules are the same used by TAS. As for the sampling rule, S-TAS defines a confidence region $C_t$ around $\hat{\boldsymbol{\mu}}(t)$, *i.e.*, $C_t = \{\boldsymbol{\lambda} \in \mathcal{M} : \sum_{k \in [K]} N_k(t) d(\hat{\mu}_k(t), \lambda_k) \leq 8K \log(t)\}$, and computes a set $\mathcal{I}_t$ of candidate answers as follows: $\mathcal{I}_t = \bigcup_{\boldsymbol{\lambda} \in C_t} i_F(\boldsymbol{\lambda})$. Then, S-TAS selects an answer $i_t \in \mathcal{I}_t$ according to some pre-specified total order over $\mathcal{I}$, and it computes $\boldsymbol{\omega}(t) \in \boldsymbol{\omega}^\star(\hat{\boldsymbol{\mu}}(t), \neg i_t)$ for the selected answer $i_t$. Finally, it selects the next action $A_t$ by applying the C-Tracking sampling rule over the sequence $\{\boldsymbol{\omega}(t)\}_t$. The main idea behind S-TAS is that, due to the upper-hemicontinuity of $\boldsymbol{\mu} \rightrightarrows i_F(\boldsymbol{\mu})$, the set $\mathcal{I}$ will eventually collapse (under a good event) to $i_F(\boldsymbol{\mu})$ for sufficiently large $t$. Then, since $i_t$ is chosen according to a pre-specified total order over $\mathcal{I}$, $i_t$ will be fixed to some $\iota \in i_F(\boldsymbol{\mu})$, and the C-Tracking sampling rule will ensure that $\inf_{\boldsymbol{\omega} \in \boldsymbol{\omega}^\star(\boldsymbol{\mu}, \neg \iota)} \|\boldsymbol{N}(t)/t - \boldsymbol{\omega}\| \to 0$. As shown by Degenne & Koolen (2019), this property leads asymptotic optimality both in single and multiple-answer problems.

## 3 NON-ASYMPTOTIC ANALYSIS FOR TRACK-AND-STOP

First, we present two assumptions that we will use in our analysis.

**Assumption 1** (Sub-Gaussian Arms). *Arms belongs to a $\sigma^2$-sub-Gaussian exponential family,* i.e., *for all $\mu, \mu' \in \Theta$, it holds $d(\mu, \mu') \geq \frac{(\mu - \mu')^2}{2\sigma^2}$.*

**Assumption 2** (Bounded parameters). *There exists $[\mu_{\min}, \mu_{\max}] \subset \Theta$ such that $\mathcal{M} \subset [\mu_{\min}, \mu_{\max}]$.*

Both Assumptions 1 and 2 are mild requirements that have been frequently adopted in the literature; see *e.g.*, Degenne et al. (2019; 2020); Jourdan et al. (2021); Poiani et al. (2024). Assumption 1 is used primarily for concentration arguments. Assumption 2 implies that, for any two distributions $p$ and $q$ within $[\mu_{\min}, \mu_{\max}]$ it holds that $d(p, q) \leq L$ and $|\nu_p - \nu_q| \leq D$, for some constants $L$ and $D$. These two properties are the main reason for introducing Assumption 2 in our analysis.[7]

Before introducing our result, we make a minor modification to the TAS algorithm that allows for a simpler analysis. Specifically, instead of computing $\boldsymbol{\omega}(t) \in \boldsymbol{\omega}^\star(\hat{\boldsymbol{\mu}}(t))$, it computes $\boldsymbol{\omega}(t) \in \boldsymbol{\omega}^\star(\tilde{\boldsymbol{\mu}}(t))$, where $\tilde{\boldsymbol{\mu}}(t)$ denotes the orthogonal projection of $\hat{\boldsymbol{\mu}}(t)$ onto $[\mu_{\min}, \mu_{\max}]^K$.[8] This modification is only required to ensure that $d(\tilde{\mu}_k(t), \cdot)$ "well-behaves" whenever $t$ is small. Such projection has already been adopted in sampling rules for algorithms that provide finite-confidence guarantees, see, *e.g.*, the regret minimization approach presented in Degenne et al. (2019). More formally, its purpose is ensuring that $d(\tilde{\mu}_k(t), \lambda)$ is bounded for all steps and any $\lambda \in [\mu_{\min}, \mu_{\max}]$. We note that this modification is only needed to handle pathological cases that might arises when dealing with arbitrary canonical exponential families and it is not needed, *e.g.*, when the family of distributions is Gaussian. Later in this section, we discuss how to drop the projection step and how this affects the resulting guarantees. We are now ready to state our finite-confidence result for TAS.

**Theorem 1** (Non-Asymptotic Bound for TAS). *Let $i^\star(\cdot)$ be single-valued, and suppose that Assumption 1 and Assumption 2 hold. Then, the expected stopping time of TAS satisfies $\mathbb{E}_{\boldsymbol{\mu}}[\tau_\delta] \leq 2eK + 10K^4 + T_0(\delta)$, where $T_0(\delta)$ is given by*

$$T_0(\delta) = \inf \left\{ t \in \mathbb{N} : \beta_{t,\delta} \leq t T^\star(\boldsymbol{\mu})^{-1} - g(t) \right\}, \tag{5}$$

---

[5]For completeness, we report a formal statement and a proof in Lemma 10.

[6]See Lemma 6 and Theorem 7 in Degenne et al. (2019).

[7]We refer the interested reader to Appendix C for further details.

[8]Note that, since $\mathcal{M}$ is know by definition, so are $\mu_{\min}$ and $\mu_{\max}$. In other words, this kind of modification does not require additional knowledge of the problem.

*where* $g(t) = 64\sigma DLK^2 \log(K)\sqrt{t \log^2(t)} + 16\sigma D\sqrt{Kt^{3/2} \log(t)}$.

Theorem 1 provides the finite-confidence bound on the performance of TAS. First, we note that the upper bound is expressed as a sum of three terms, *i.e.*, $2eK$, $10K^4$ and $T_0(\delta)$. The first two $\delta$-independent terms are artifact of the analysis and their origin is detailed in the proof sketch provided below. The last and more important term, $T_0(\delta)$, is a function of $\delta$, which essentially captures how quickly the quantity $\max_{i \in \mathcal{I}} \inf_{\boldsymbol{\lambda} \in \neg i} \sum_{k \in [K]} N_k(t)d(\hat{\mu}_k(t), \lambda_k)$ is approaching the stopping threshold $\beta_{t,\delta}$. Indeed, $tT^\star(\boldsymbol{\mu})^{-1} - g(t)$ is essentially a lower bound (under a good event) on the aforementioned optimization problem: when this quantity exceeds $\beta_{t,\delta}$, TAS stops. In other words, $T_0(\delta)$ measures how fast TAS is gathering information to discriminate $i^\star(\boldsymbol{\mu})$ from all the other candidate answers. By re-arranging the condition in Equation (5), *i.e.*, $\beta_{t,\delta} + g(t) \leq tT^\star(\boldsymbol{\mu})^{-1}$, we can see that the r.h.s. grows linearly in $t$, while the l.h.s. is growing sub-linearly with a rate of $\mathcal{O}(\log(1/\delta) + t^{3/4})$.[9] This ensures that $T_0(\delta)$ is finite and that Theorem 1 recovers the asymptotic optimality of TAS whenever $\delta \to 0$. Finally, although $T_0(\delta)$ is defined somehow implicitly, in Appendix G we derive a further upper bound that highlights that the scaling is $T^\star(\boldsymbol{\mu}) \log(1/\delta)$ up to polylogarithmic factors and constant terms.

**Proof Sketch** As we discussed, the original analysis of TAS is asymptotic in nature. In contrast, we follow a different path which is inspired by finite-confidence analysis in the literature, *e.g.*, Degenne et al. (2019); Jourdan & Degenne (2023). In particular, we conduct the analysis under a sequence of good events $\{\mathcal{E}_t\}_t$. Specifically, we consider $\mathcal{E}_t = \left\{ \forall s \in \left[\lceil\sqrt{t}\rceil, t\right] : \sum_{k \in [K]} N_k(s)d(\hat{\mu}_k(s), \mu_k) \leq 8K \log(s) \right\}$. This sequence of events has two desirable properties. First, one can show that $\sum_{t=3}^{+\infty} \mathbb{P}_{\boldsymbol{\mu}}(\mathcal{E}_t^c) \leq 2eK$ (see Lemma 9). Second, as we discuss below, there exists a time $\bar{T}$ such that for all $t \geq \bar{T}$, $\mathcal{E}_t$ implies stopping, namely $\mathcal{E}_t \subseteq \{\tau_\delta \leq t\}$. Using these two properties one obtains that $\mathbb{E}_{\boldsymbol{\mu}}[\tau_\delta] \leq \bar{T} + 2eK$ (see Lemma 1). In the remainder of the proof, we will show how $\bar{T} := T_0(\delta) + 10K^4$ satisfies the requirement mentioned earlier. Before doing that, we introduce some additional notation. Recall that $\boldsymbol{\omega}(s) \in \operatorname{argmax}_{\boldsymbol{\omega} \in \Delta_K} \max_{i \in \mathcal{I}} \inf_{\boldsymbol{\lambda} \in \neg i} \sum_{k \in [K]} \omega_k d(\tilde{\mu}_k(s), \lambda_k)$. Then, we denote by $i_s \in \mathcal{I}$ any answer that attains the argmax when paired with $\boldsymbol{\omega}(s)$.

Now, the key idea is analyzing the stopping rule of TAS and, in particular, lower bounding $\max_{i \in \mathcal{I}} \inf_{\boldsymbol{\lambda} \in \neg i} \sum_{k \in [K]} N_k(t)d(\hat{\mu}_k(t), \lambda_k)$ to obtain $tT^\star(\boldsymbol{\mu})^{-1} - g(t)$. To this end, as we shall see, the crucial step is approximating (up to a sublinear in $t$ factor) the max-min problem of the stopping rule with what TAS uses in the sampling rule in each round $s \geq \sqrt{t}$, *i.e.*, $\inf_{\boldsymbol{\lambda} \in \neg i_s} \sum_{k \in [K]} \omega_k(s)d(\tilde{\mu}_k(s), \lambda_k)$. Further comments on this are provided right after the proof sketch. Now, for any $t \geq 10K^4$,[10] if TAS has not stopped at time $t$, then the following holds:

$$\beta_{t,\delta} > \inf_{\boldsymbol{\lambda} \in \neg i^\star(\boldsymbol{\mu})} \sum_{k \in [K]} N_k(t)d(\hat{\mu}_k, \lambda_k) \qquad \text{(Stopping Rule)}$$

$$\gtrsim \sum_{s=1}^{t} \inf_{\boldsymbol{\lambda} \in \neg i^\star(\boldsymbol{\mu})} \sum_{k \in [K]} \omega_k(s)d(\mu_k, \lambda_k) - \widetilde{\mathcal{O}}(\sqrt{t}) \qquad (\mathcal{E}_t + \text{C-Tracking})$$

$$\geq \sum_{s=1}^{t} \inf_{\boldsymbol{\lambda} \in \neg i_s} \sum_{k \in [K]} \omega_k(s)d(\mu_k, \lambda_k) - \widetilde{\mathcal{O}}(\sqrt{t}).$$

Here, C-tracking ensures that $\boldsymbol{N}(t) \approx \sum_{s=1}^{t} \boldsymbol{\omega}(s)$, and under the event $\mathcal{E}_t$ we have $d(\hat{\mu}_k(t), \cdot) \approx d(\mu_k, \cdot)$. In the last step, we have used that if $i_s = i^\star(\boldsymbol{\mu})$ then the claim is trivial, and if $i_s \neq i^\star(\boldsymbol{\mu})$, then, $\boldsymbol{\mu} \in \neg i_s$. We observe that this argument explicitly relies on the fact that $i^\star(\boldsymbol{\mu})$ is single-valued.[11] Now, we analyze the information accumulated by TAS by lower bounding $\sum_{s=1}^{t} \inf_{\boldsymbol{\lambda} \in \neg i_s} \sum_{k \in [K]} \omega_k(s)d(\mu_k, \lambda_k)$. Using the definition of $\mathcal{E}_t$ and the fact that $\tilde{\mu}_k(s) \in$

---

[9] Here, we plugged in $\beta_{t,\delta} \approx \log(1/\delta) + K\log(t)$.

[10] This requirement is due to some technical step that is used at the end of the proof.

[11] This is important to be noted, otherwise it might seems that TAS achieves asymptotically optimal results even in problems with multiple correct answers.

$[\mu_{\min}, \mu_{\max}]$,[12] we have that

$$\sum_{s=1}^{t} \inf_{\boldsymbol{\lambda} \in \neg i_s} \sum_{k \in [K]} \omega_k(s) d(\mu_k, \lambda_k) \gtrsim \sum_{s \geq \sqrt{t}} \inf_{\boldsymbol{\lambda} \in \neg i_s} \sum_{k \in [K]} \omega_k(s) d(\tilde{\mu}_k(s), \lambda_k) - \widetilde{\mathcal{O}}(\sqrt{t}).$$

We have reached our goal of lower bounding the stopping rule of TAS with its sampling rule. Using the definition of $i_s$ and $\boldsymbol{\omega}(s)$, this allows for the following inequalities:

$$\sum_{s \geq \sqrt{t}} \inf_{\boldsymbol{\lambda} \in \neg i_s} \sum_{k \in [K]} \omega_k(s) d(\tilde{\mu}_k(s), \lambda_k) = \sum_{s \geq \sqrt{t}} \sup_{\boldsymbol{\omega} \in \Delta_K} \max_{i \in \mathcal{I}} \inf_{\boldsymbol{\lambda} \in \neg i} \sum_{k \in [K]} \omega_k(s) d(\tilde{\mu}_k(s), \lambda_k)$$

$$\geq \sum_{s \geq \sqrt{t}} \inf_{\boldsymbol{\lambda} \in \neg i^\star(\boldsymbol{\mu})} \sum_{k \in [K]} \omega_k^\star d(\tilde{\mu}_k(s), \lambda_k) \quad \text{(for } \boldsymbol{\omega}^\star \in \boldsymbol{\omega}^\star(\boldsymbol{\mu}))$$

$$= (t - \sqrt{t} - 1) T^\star(\boldsymbol{\mu})^{-1} - \widetilde{\mathcal{O}}(\sqrt{t^{3/2}}), \quad \text{(By } \mathcal{E}_t)$$

where the second step holds for any $\boldsymbol{\omega}^\star \in \boldsymbol{\omega}^\star(\boldsymbol{\mu})$ and the last one requires an algebraic step that requires $t \geq 10K^4$. Intuitively, however, this last step is still using the fact that $d(\tilde{\mu}_k(s), \cdot) \approx d(\mu_k, \cdot)$ under the good event. Chaining together all the terms within the $\widetilde{O}(\cdot)$ yields the desired result.

**The proof idea** As anticipated above, the main idea is approximating up to a sub-linear in $t$ factor, the condition used in the stopping rule with the quantity $\sum_{s \geq \sqrt{t}} \inf_{\boldsymbol{\lambda} \in \neg i_s} \sum_{k \in [K]} \omega_k(s) d(\tilde{\mu}_k(s), \lambda_k)$, which is what TAS uses in its sampling rule. Once this is done, we can use the definition of $\boldsymbol{\omega}(s)$ and $i_s$ to introduce the optimal weights $\boldsymbol{\omega}^\star$ for the underlying unknown problem and the infimum over $\neg i^\star(\boldsymbol{\mu})$. Importantly, we observe that the generalization of $\boldsymbol{\omega}^\star$ that we provided in Equation (2) played a crucial role. Finally, by upper-bounding the difference between $d(\tilde{\mu}_k(s), \lambda_k)$ and $d(\mu_k, \lambda_k)$, we introduce $T^\star(\boldsymbol{\mu})^{-1}$, which is the desired quantity as it allows to recover the asymptotic optimality.

**Removing the projection step** We discuss how to obtain finite-confidence guarantees for a version of TAS that does not use projection in the sampling rule, *i.e.*, exactly the version of TAS by Garivier & Kaufmann (2016). Before that, we make a remark on Assumption 2. Let $\boldsymbol{\mu} \in \mathcal{M}$ and let $F_k = \min\{|\mu_k - \mu_{\min}|, |\mu_k - \mu_{\max}|\}$ and $F = \min_{k \in [K]} F_k$. Then, since $\Theta$ is an open interval and since $[\mu_{\min}, \mu_{\max}]$ is closed, it follows that $F_k > 0 \; \forall k \in [K]$, and thus $F > 0$. That being said, the simplest way to analyze TAS without projection follows by noticing that there exists a time $T_\mathcal{M} \in \mathbb{N}$ such that, for all $t \geq T_\mathcal{M}$, on $\mathcal{E}_t$, it holds that $\hat{\boldsymbol{\mu}}(s) \in [\mu_{\min}, \mu_{\max}]$ for all $s \geq \sqrt{t}$ (see Lemma 3 in Appendix E). $T_\mathcal{M}$ depends only $\mathcal{M}$ and its distance $F$ from the interval $[\mu_{\min}, \mu_{\max}]$; precisely:

$$T_\mathcal{M} = \max\left\{10K^4, \inf\left\{n \in \mathbb{N} : \sqrt{\frac{64\sigma^2 K \log(n)}{\sqrt{\sqrt{n} + K^2} - 2K}} \leq F\right\}\right\}. \tag{6}$$

This allows us analyze the stopping time under a good event in the same way that we did above. Indeed, it is sufficient that $d(\hat{\mu}_k(s), \cdot)$ well-behaves only at steps $s \geq \sqrt{t}$. Thus, the only difference with respect to Theorem 1 would be the additional term $T_\mathcal{M}$ in the upper bound of $\mathbb{E}_{\boldsymbol{\mu}}[\tau_\delta]$.

## 4 NON-ASYMPTOTIC ANALYSIS FOR STICKY TRACK-AND-STOP

In this section, we present the finite-confidence analysis of S-TAS. As for TAS, we will rely on Assumptions 1 and 2. Furthermore, for reasons similar to those discussed above, we consider a slightly modified version of S-TAS that incorporates a projection into its sampling rule. Specifically, the algorithm computes $\boldsymbol{\omega}(s) \in \boldsymbol{\omega}^\star(\tilde{\boldsymbol{\mu}}(s), \neg i_s)$.[13] The following theorem summarizes our result.

---

[12]This is needed to upper bound $d(\tilde{\mu}_k(s), \lambda_k) - d(\mu_k, \lambda_k)$. Indeed, $d(\tilde{\mu}_k(s), \lambda_k) - d(\mu_k, \lambda_k) \leq (\nu_{\tilde{\mu}_k(s)} - \nu_{\lambda_k})|\tilde{\mu}_k(s) - \mu_k| \leq D|\tilde{\mu}_k(s) - \mu_k|$ since $\tilde{\boldsymbol{\mu}}(s), \boldsymbol{\lambda} \in [\mu_{\min}, \mu_{\max}]$.

[13]Using the same argument that we discussed in Section 3, it is possible to analyze the version of S-TAS that does not use projection. The sample complexity results differ only by the additional term $T_\mathcal{M}$.

**Theorem 2** (Non-Asymptotic Bound for Sticky-TAS). *Suppose that Assumption 1 and Assumption 2 hold. Let $\epsilon_{\boldsymbol{\mu}} > 0$ be any number such that, for all $\boldsymbol{\mu}' : \|\boldsymbol{\mu} - \boldsymbol{\mu}'\|_\infty \leq \epsilon_{\boldsymbol{\mu}}$, it holds that $i_F(\boldsymbol{\mu}') \subseteq i_F(\boldsymbol{\mu}) \cup (\mathcal{I} \setminus i^\star(\boldsymbol{\mu}))$, and let $T_{\boldsymbol{\mu}} \in \mathbb{N}$ be defined as follows:*

$$T_{\boldsymbol{\mu}} = \max\left\{10K^4, \inf\left\{n \in \mathbb{N} : \sqrt{\frac{64K\sigma^2 \log(n)}{\sqrt{\sqrt{n} + K^2} - 2K}}\right\} \leq \epsilon_{\boldsymbol{\mu}}\right\}.$$

*Then, it holds that $\mathbb{E}_{\boldsymbol{\mu}}[\tau_\delta] \leq 2eK + 10K^4 + T_0(\delta)$, where $T_0(\delta)$ is given by*

$$T_0(\delta) = \inf\left\{t \in \mathbb{N} : \beta_{t,\delta} \leq (t - T_{\boldsymbol{\mu}}) T^\star(\boldsymbol{\mu})^{-1} - g(t)\right\},$$

*where $g(t) = 80\sigma DLK^2 \log(K)\sqrt{t \log^2(t)} + 32\sigma D\sqrt{Kt^{3/2} \log(t)}$.*

Theorem 2 provides a finite-confidence bound for S-TAS in multiple-answer problems. As one can notice, the result is similar in nature to what we presented for TAS in Theorem 1. In particular, the expression of $T_0(\delta)$ is similar to that of TAS, and, for the same reasons outlined in Section 3, this allows us to recover the asymptotic optimality guarantees of Degenne & Koolen (2019) whenever $\delta \to 0$.[14] The main difference between Theorem 1 and Theorem 2 is the presence of an additional problem-dependent constant $T_{\boldsymbol{\mu}} T^\star(\boldsymbol{\mu})^{-1}$ within the expression of $T_0(\delta)$. As our proof will reveal, $T_{\boldsymbol{\mu}}$ is the time that is needed by S-TAS (under the good event) to distinguish $i_F(\boldsymbol{\mu}) \cup (\mathcal{I} \setminus i^\star(\boldsymbol{\mu}))$ from $i^\star(\boldsymbol{\mu}) \setminus i_F(\boldsymbol{\mu})$. In other words, from that point on, under $\mathcal{E}_t$, all the candidate models $\boldsymbol{\mu}'$ within the confidence region $C_s$ satisfy $i_F(\boldsymbol{\mu}') \subseteq i_F(\boldsymbol{\mu}) \cup (\mathcal{I} \setminus i^\star(\boldsymbol{\mu}))$ for all $s \geq \sqrt{t}$. Indeed, whenever $t \geq T_{\boldsymbol{\mu}}$ it will be possible to link the stopping rule to $T^\star(\boldsymbol{\mu})^{-1}$ (with some sub-linear terms) as we did for TAS.[15]

Finally, we comment on the nature of $T_{\boldsymbol{\mu}}$. In particular, the existence of $\epsilon_{\boldsymbol{\mu}} > 0$ is guaranteed by the upper hemicontinuity of the set-valued function $i^\star(\boldsymbol{\mu})$. Our claim holds for any $\epsilon_{\boldsymbol{\mu}}$ that satisfies $\forall \boldsymbol{\mu}' : \|\boldsymbol{\mu} - \boldsymbol{\mu}'\|_\infty \leq \epsilon_{\boldsymbol{\mu}} \implies i_F(\boldsymbol{\mu}') \subseteq i_F(\boldsymbol{\mu}) \cup (\mathcal{I} \setminus i^\star(\boldsymbol{\mu}))$, and, consequently, the tightest bound is obtained for the largest possible $\epsilon_{\boldsymbol{\mu}}$. To conclude, we observe that the definition of $\epsilon_{\boldsymbol{\mu}}$ is intrinsic to the definition of the task at hand, *i.e.*, $i^\star(\boldsymbol{\mu})$. Once the task is fixed, it might be possible to obtain a more explicit characterization of this quantity. Consider, for instance, the relevant case of an $\epsilon$-best arm identification problem. Here, it holds that $i_F(\boldsymbol{\mu}) = \text{argmax}_{i \in [K]} \mu_i$ (see, *e.g.*, Garivier & Kaufmann, 2021; Jourdan et al., 2023). Let $\Delta_{\boldsymbol{\mu}} = \min_{i \notin i_F(\boldsymbol{\mu})} \mu_\star - \mu_i$ where $\mu_\star$ is the value of any optimal arm. Then, whenever $\epsilon_{\boldsymbol{\mu}} < \Delta_{\boldsymbol{\mu}}$, we have that $i_F(\boldsymbol{\mu}') \subseteq i_F(\boldsymbol{\mu})$. Hence, Theorem 2 holds, for instance, with $\epsilon_{\boldsymbol{\mu}} = \Delta_{\boldsymbol{\mu}}/2$.

Now, we present a proof sketch of the result.

**Proof Sketch** As we did for TAS, the analysis is carried out under a sequence of good events $\{\mathcal{E}_t\}_t$ which are exactly the ones that we considered when proving Theorem 1. As above, we will show that for $\bar{T} = 10K^4 + T_0(\delta)$ and $t \geq \bar{T}$, we have that $\mathcal{E}_t \subseteq \{\tau_\delta \leq t\}$. As a consequence, $\mathbb{E}_{\boldsymbol{\mu}}[\tau_\delta] \leq 2eK + 10K^4 + T_0(\delta)$. To do this, the main idea is lower bounding the condition used in the stopping rule with what S-TAS uses in the sampling rule. First, we state an intermediate result, which is a consequence of (i) the forced exploration of S-TAS, (ii) the definition of the region $C_t$ of candidate models, and (iii) the upper hemicontinuity of the set-valued function $i_F(\boldsymbol{\mu})$. Specifically, in Lemma 4 we prove that, $\forall t \geq T_{\boldsymbol{\mu}}$, on $\mathcal{E}_t$, it holds that:

$$i_F(\boldsymbol{\mu}') \subseteq i_F(\boldsymbol{\mu}) \cup (\mathcal{I} \setminus i^\star(\boldsymbol{\mu})) \quad \forall s \geq \sqrt{t} \text{ and } \boldsymbol{\mu}' \in C_s. \tag{7}$$

Indeed, by upper hemicontinuity, models $\boldsymbol{\mu}'$ similar to $\boldsymbol{\mu}$ have answers in $i_F(\boldsymbol{\mu}')$ which are "close" to the ones in $i_F(\boldsymbol{\mu})$, and models in $C_t$ shrink toward $\boldsymbol{\mu}$ due to the forced exploration of the algorithm.

We now analyze the amount of information that is gathered by S-TAS under the good event $\mathcal{E}_t$. Let $t \geq 10K^4$ and let $\widetilde{T} = \max\{\lceil\sqrt{t}\rceil, T_{\boldsymbol{\mu}}\}$. Denote by $\imath \in \mathcal{I}$, the answer that is selected from $i_F(\boldsymbol{\mu})$ by the pre-specified total order over $\mathcal{I}$. Then, for $t \geq 10K^4$, if S-TAS has not stopped at $t$, we have

---

[14]As we did for TAS, in Appendix G we provide an explicit upper bound on $T_0(\delta)$.

[15]It is interesting to note that in our proof we are not using $\mathcal{I}_s = i_F(\boldsymbol{\mu})$, *i.e.*, that S-TAS has actually "sticked" to an answer in $i_F(\boldsymbol{\mu})$. Instead, it is sufficient that $\mathcal{I}_s$ excludes answers in $i^\star(\boldsymbol{\mu})$ that are not within $i_F(\boldsymbol{\mu})$.

that:

$$\beta_{t,\delta} \gtrsim \sum_{s=1}^{t} \inf_{\boldsymbol{\lambda} \in \neg \imath} \sum_{k \in [K]} \omega_k(s) d(\mu_k, \lambda_k) - \widetilde{\mathcal{O}}(\sqrt{t}) \tag{8}$$

$$\geq \sum_{s=\widetilde{T}}^{t} \inf_{\boldsymbol{\lambda} \in \neg i_s} \sum_{k \in [K]} \omega_k(s) d(\mu_k, \lambda_k) - \widetilde{\mathcal{O}}(\sqrt{t}), \tag{9}$$

where the first step is due to concentration and C-Tracking, and the second one uses Equation (7). Indeed, for $s \geq \widetilde{T}$, either $i_s = \imath$ (and in this case the claim is trivial), or $i_s \neq \imath$. In this second case, from Equation (7) we have that $i_s \notin i^\star(\boldsymbol{\mu})$ and, therefore, $\boldsymbol{\mu} \in \neg i_s$. Now, by concentration arguments (*i.e.*, $d(\tilde{\mu}_k(s), \cdot) \approx d(\mu_k, \cdot)$), and using the definition of $\boldsymbol{\omega}(s)$, we have that:

$$\sum_{s=\widetilde{T}}^{t} \inf_{\boldsymbol{\lambda} \in \neg i_s} \sum_{k \in [K]} \omega_k(s) d(\mu_k, \lambda_k) \gtrsim \sum_{s=\widetilde{T}}^{t} \inf_{\boldsymbol{\lambda} \in \neg i_s} \sum_{k \in [K]} \omega_k(s) d(\tilde{\mu}_k(s), \lambda_k) - \widetilde{\mathcal{O}}(\sqrt{t})$$

$$= \sum_{s=\widetilde{T}}^{t} \max_{\boldsymbol{\omega} \in \Delta_K} \inf_{\boldsymbol{\lambda} \in \neg i_s} \sum_{k \in [K]} \omega_k d(\tilde{\mu}_k(s), \lambda_k) - \widetilde{\mathcal{O}}(\sqrt{t}).$$

The next step is crucial for relating the amount of gathered information to $T^\star(\boldsymbol{\mu})^{-1}$. Let $\boldsymbol{\mu}'(s) \in C_s$ be such that $i_s \in i_F(\boldsymbol{\mu}'(s))$. From concentration arguments and the definition of $\boldsymbol{\mu}'(s)$, we have that:

$$\sum_{s=\widetilde{T}}^{t} \max_{\boldsymbol{\omega} \in \Delta_K} \inf_{\boldsymbol{\lambda} \in \neg i_s} \sum_{k \in [K]} \omega_k d(\tilde{\mu}_k(s), \lambda_k) \gtrsim \sum_{s=\widetilde{T}}^{t} \max_{\boldsymbol{\omega} \in \Delta_K} \inf_{\boldsymbol{\lambda} \in \neg i_s} \sum_{k \in [K]} \omega_k d(\mu'_k(s), \lambda_k) - \widetilde{\mathcal{O}}\left(\sqrt{t^{3/2}}\right)$$

$$= \sum_{s=\widetilde{T}}^{t} \max_{\boldsymbol{\omega} \in \Delta_K} \max_{i \in \mathcal{I}} \inf_{\boldsymbol{\lambda} \in \neg i} \sum_{k \in [K]} \omega_k d(\mu'_k(s), \lambda_k) - \widetilde{\mathcal{O}}\left(\sqrt{t^{3/2}}\right)$$

$$\geq \sum_{s=\widetilde{T}}^{t} \inf_{\boldsymbol{\lambda} \in \neg \imath} \sum_{k \in [K]} \omega_k^\star d(\mu'_k(s), \lambda_k) - \widetilde{\mathcal{O}}\left(\sqrt{t^{3/2}}\right),$$

for any $\boldsymbol{\omega}^\star \in \boldsymbol{\omega}^\star(\boldsymbol{\mu}, \neg \imath)$. The first step follows by observing that $d(\tilde{\mu}_k(s), \cdot)$ can be upper-bounded by $d(\hat{\mu}_k(s), \cdot)$, and $d(\hat{\mu}_k(s), \cdot) \approx d(\mu'(s), \cdot)$ since $\boldsymbol{\mu}'(s) \in C_s$ by definition. Then, the proof is simply concluded by noticing that:

$$\sum_{s=\widetilde{T}}^{t} \inf_{\boldsymbol{\lambda} \in \neg \imath} \sum_{k \in [K]} \omega_k^\star d(\mu'_k(s), \lambda_k) \gtrsim \sum_{s=\widetilde{T}}^{t} \inf_{\boldsymbol{\lambda} \in \neg \imath} \sum_{k \in [K]} \omega_k^\star d(\mu_k, \lambda_k) - \widetilde{\mathcal{O}}(\sqrt{t^{3/2}})$$

$$= (T - \widetilde{T}) T^\star(\boldsymbol{\mu})^{-1} - \widetilde{\mathcal{O}}(\sqrt{t^{3/2}}),$$

where the first step follows from concentration arguments and the fact that $\boldsymbol{\mu}'(s) \in C_s$. Rearrenging all the terms yields the desired result.

**The proof idea**  As for TAS, the general idea is approximating with sub-linear terms the stopping rule with $\sum_{s=\widetilde{T}}^{t} \inf_{\boldsymbol{\lambda} \in \neg i_s} \sum_{k \in [K]} \omega_k(s) d(\tilde{\mu}_k(s), \lambda_k)$, that is what S-TAS uses in its sampling rule. Now, there are two key differences with respect to Theorem 1. First, to reach such objective we need to consider sufficiently large timesteps, *i.e.*, $s \geq \widetilde{T}$. The issue is that when $s$ is small, the S-TAS sampling rule has no control over the selected answers $i_s$ (apart from a generic total order over $\mathcal{I}$). This does not allow to easily switch from $\neg \imath$ to $\neg i_s$, *i.e.*, the step from Equation (8) to Equation (9). Second, once we have obtained $\sum_{s=\widetilde{T}}^{t} \inf_{\boldsymbol{\lambda} \in \neg i_s} \sum_{k \in [K]} \omega_k(s) d(\tilde{\mu}_k(s), \lambda_k)$, this does not allow us to directly introduce $\boldsymbol{\omega}^\star(\boldsymbol{\mu}, \neg \imath)$ and $\neg \imath$ as we did for TAS. An intermediate step is necessary. This requires studying the difference between $d(\tilde{\mu}_k(s), \cdot)$ and $d(\mu'_k(s), \cdot)$. The reason is that $i_s$ is an answer that attains the argmax only when paired with a model $\boldsymbol{\mu}'(s) \in C_s$ such that $i_s \in i_F(\boldsymbol{\mu}'(s))$.

**S-TAS in single-answer problems**   Whenever $i^\star(\boldsymbol{\mu})$ is single-valued, the dependency on $T_{\boldsymbol{\mu}}$ can be removed as the step from Equation (8) and Equation (9) follows directly from the fact that $|i^\star(\boldsymbol{\mu})| = 1$ (as we did for TAS). Thus, one would obtain a result identical to Theorem 1 (*i.e.*, the same bound up to constant multiplicative terms). Nonetheless, we actually note that the two proofs are still different, and the reason is the different sampling rules adopted by the two algorithms. Specifically, in S-TAS $i_s$ is an answer in $i_F(\boldsymbol{\mu}'(s))$ for some $\boldsymbol{\mu}'(s) \in C_s$ and $\boldsymbol{\omega}(s) \in \boldsymbol{\omega}^\star(\tilde{\boldsymbol{\mu}}(s), \neg i_s)$. On the other hand, TAS directly uses $\boldsymbol{\omega}(s) \in \boldsymbol{\omega}^\star(\tilde{\boldsymbol{\mu}}(s), \neg i_s)$ for $i_s \in i_F(\tilde{\boldsymbol{\mu}}(s))$.

**On the behavior of S-TAS**   It is interesting to observe that our proof differs significantly from the one of asymptotic optimality by Degenne & Koolen (2019). Beyond the obvious distinction (*i.e.*, our analysis is non-asymptotic), we also note that the proof of Theorem 2 does not rely on what $\boldsymbol{N}(t)/t$ converges to, nor does it exploit the convexity of the set $\boldsymbol{\omega}(\boldsymbol{\mu}, \neg\imath)$, which instead were key components in the analysis of Degenne & Koolen (2019). Instead, we only reason in terms of "information" collected by S-TAS by analyzing *values* of functions of the form $\sum_{k\in[K]} \omega_k d(\cdot, \cdot)$.

## 5   CONCLUSION

This work provided the first finite-confidence characterization of the performance of Track-and-Stop and Sticky Track-and-Stop, two general algorithms that are able to solve optimally a large spectrum of pure exploration settings. Overall, we solve two open problems in the literature. First, Theorem 1 sheds light on the finite-confidence guarantees of TAS, thus providing theoretical support on why the algorithm usually enjoys good performance in practice. Secondly, Theorem 2 provides the first finite-confidence guarantees for the general multiple-answer setting. To conclude, we note that our results (Theorem 1 and Theorem 2) have simple and natural proofs, and they both recover the asymptotic optimality whenever $\delta$ goes to 0.

Several questions remain open. For instance, is it possible to improve the finite-confidence analysis of Sticky Track-and-Stop by removing the presence of the problem constant $T_{\boldsymbol{\mu}}$? We conjecture that this would require to slightly modify the sampling rule. Indeed, by selecting answers $i_t$ more strategically than using any total order over $\mathcal{I}$ might lead to stronger finite-confidence results and, eventually, more competitive performance. Furthermore, there remains a gap between lower and upper bounds in the finite-confidence regime (Degenne et al., 2019; Wang et al., 2021; Barrier et al., 2022; Jourdan & Degenne, 2023; Jourdan et al., 2023). Future work should focus on analyzing in greater details the finite-confidence regime of general pure exploration problems. Here, one could draw inspiration from the works of Simchowitz et al. (2017); Al Marjani et al. (2022); Poiani et al. (2024), where the authors shed some light on the challenges in developing lower bounds and algorithms that enjoy tight dependencies not only in $\log(1/\delta)$, but also in other parameters of the instance (*e.g.*, $K$). Finally, Poiani et al. (2025) recently extended TAS and S-TAS to problems where the set of correct answers is possibly infinite. However, their algorithm only attains asymptotic optimality, and our analysis cannot be straightforwardly extended to this setting. Future works, should focus on closing this gap. Here, it is relevant to mention the recent work of Osogami et al. (2025), where the authors proposed an algorithm that enjoys finite confidence guarantees in a specific infinite answer problem, *i.e.*, that of estimating, up to an accuracy $\epsilon$, the value of the optimal arm.

ACKNOWLEDGMENTS

Funded by the European Union. Views and opinions expressed are however those of the author(s) only and do not necessarily reflect those of the European Union or the European Research Council Executive Agency. Neither the European Union nor the granting authority can be held responsible for them.

Work supported by the Cariplo CRYPTONOMEX grant and by an ERC grant (Project 101165466 — PLA-STEER).

REPRODUCIBILITY STATEMENT

The nature of this work is theoretical. We precisely stated and discussed the assumptions that are required to derive our result in the main text (see Assumptions 1 and 2), and we further discussed the assumptions in Appendix C. In the main text, we provided proof sketches for both Theorem 1 and Theorem 2, and we included complete proofs in Appendix D and Appendix F.

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

CONTENTS

# A STRUCTURED BANDITS

For completeness, we now show that how to encode known structure such as Lipschitzianity and unimodality through $\mathcal{M}$.

We first focus on the the best-arm identification problem in Lipshitz bandit with finite arms, *i.e.*, the same setting studied in (Wang et al., 2021). This structure can be formalized as follows:

$$\mathcal{M} = \{\boldsymbol{\mu} \in [\mu_{\min}, \mu_{\max}]^K : \exists i \; s.t. \; \mu_i > \mu_k, \wedge \; \forall k, k', |\mu_k - \mu_{k'}| \leq \|a_k - a_{k'}\|_\infty\},$$

where $l > 0$ is a known constant, $a_k \in \mathbb{R}^d$ are the known feature vectors for the arms, and $[\mu_{\min}, \mu_{\max}]$ are the boundary parameters that we introduced in Assumption 2.

Next, we consider the unimodal setting of Poiani et al. (2024). In this case, we have that:

$$\mathcal{M} = \{\boldsymbol{\mu} \in [\mu_{\min}, \mu_{\max}]^K : \exists i \in [K] : \mu_i > \mu_{i+1} \geq \cdots \geq \mu_K \wedge \mu_i > \mu_{i-1} \geq \cdots \geq \mu_1\}.$$

In other words, each bandit is characterized by an unknown index $i$ such that, the arms' mean will consistently decrease both after and before $i$.

This reasoning can also be extended to other structures such as the dueling bandit formulation of Wang et al. (2021).

# B LOWER BOUND

## B.1 SINGLE-ANSWER PROBLEMS

In this section, we provide a formal statement of the lower bound for single-answer problems.

The following result follows the same arguments of Theorem 1 in Garivier & Kaufmann (2016). Since we provide two different expressions for $T^\star(\boldsymbol{\mu})$, we also report a proof for completeness.

**Proposition 1** (Lower Bound for Single-Answer Problems Garivier & Kaufmann (2016))**.** *Suppose that $|i^\star(\boldsymbol{\mu})| = 1$ for all $\boldsymbol{\mu} \in \mathcal{M}$. Let $\delta < 0.15$. For any $\boldsymbol{\mu} \in \mathcal{M}$ and any $\delta$-correct algorithm, it holds that $\mathbb{E}_{\boldsymbol{\mu}}[\tau_\delta] \geq T^\star(\boldsymbol{\mu}) \log(1/(2.4\delta))$.*

*Proof.* Let $\boldsymbol{\mu} \in \mathcal{M}$ and $\boldsymbol{\lambda} \in \neg i^\star(\boldsymbol{\mu})$. Then, from change of distribution arguments (*i.e.*, Lemma 1 in Kaufmann et al. (2016)) and the $\delta$-correctness of the algorithm, we have that:

$$\sum_{k \in [K]} \mathbb{E}_{\boldsymbol{\mu}}[N_k(\tau_\delta)] d(\mu_k, \lambda_k) \geq \log(1/(2.4\delta)).$$

Applying this result for all $\boldsymbol{\lambda} \in \neg i^\star(\boldsymbol{\mu})$ and since $\boldsymbol{\mu} \notin \mathrm{cl}(\neg i^\star(\boldsymbol{\mu}))$, we have that:

$$\log(1/(2.4\delta)) \leq \inf_{\boldsymbol{\lambda} \in \neg i^\star(\boldsymbol{\mu})} \sum_{k \in [K]} \mathbb{E}_{\boldsymbol{\mu}}[N_k(\tau_\delta)] d(\mu_k, \lambda_k)$$

$$= \mathbb{E}_{\boldsymbol{\mu}}[\tau_\delta] \inf_{\boldsymbol{\lambda} \in \neg i^\star(\boldsymbol{\mu})} \sum_{k \in [K]} \frac{\mathbb{E}_{\boldsymbol{\mu}}[N_k(\tau_\delta)]}{\mathbb{E}_{\boldsymbol{\mu}}[\tau_\delta]} d(\mu_k, \lambda_k)$$

$$\leq \mathbb{E}_{\boldsymbol{\mu}}[\tau_\delta] \sup_{\boldsymbol{\omega} \in \Delta_K} \inf_{\boldsymbol{\lambda} \in \neg i^\star(\boldsymbol{\mu})} \sum_{k \in [K]} \omega_k d(\mu_k, \lambda_k)$$

$$= \mathbb{E}_{\boldsymbol{\mu}}[\tau_\delta] \sup_{\boldsymbol{\omega} \in \Delta_K} \max_{i \in \mathcal{I}} \inf_{\boldsymbol{\lambda} \in \neg i} \sum_{k \in [K]} \omega_k d(\mu_k, \lambda_k),$$

where, the last step follows from the fact that, for all $i \neq i^\star(\boldsymbol{\mu})$, $\boldsymbol{\mu} \in \neg i$, and, hence $\inf_{\boldsymbol{\lambda} \in \neg i} \sum_{k \in [K]} \omega_k d(\mu_k, \lambda_k) = 0$. The proof then follows by the definition of $T^\star(\boldsymbol{\mu})$ together with the fact that $\sup_{\boldsymbol{\omega} \in \Delta_K} \inf_{\boldsymbol{\lambda} \in \neg i^\star(\boldsymbol{\mu})} \sum_{k \in [K]} \omega_k d(\mu_k, \lambda_k) > 0$ since $\boldsymbol{\mu} \notin \mathrm{cl}(\boldsymbol{\mu})$. $\square$

## B.2 MULTIPLE-ANSWER PROBLEMS

In this section, we formally state the lower bound for multiple answer problems.

**Proposition 2** (Lower Bound for Multiple-Answer Problems Degenne & Koolen (2019)). *Let $\mathcal{I}$ be a finite set and let $\boldsymbol{\mu} \in \mathcal{M}$. Then, it holds that:*

$$\liminf_{\delta \to 0} \frac{\mathbb{E}_{\boldsymbol{\mu}}[\tau_\delta]}{\log(1/\delta)} \geq T^\star(\boldsymbol{\mu}), \tag{10}$$

*where $T^\star(\boldsymbol{\mu})^{-1}$ is given by:*

$$T^\star(\boldsymbol{\mu})^{-1} = \sup_{\boldsymbol{\omega} \in \Delta_K} \max_{i \in i^\star(\boldsymbol{\mu})} \inf_{\boldsymbol{\lambda} \in \neg i} \sum_{k \in [K]} \omega_k d(\mu_k, \lambda_k) \tag{11}$$

$$= \sup_{\boldsymbol{\omega} \in \Delta_K} \max_{i \in \mathcal{I}} \inf_{\boldsymbol{\lambda} \in \neg i} \sum_{k \in [K]} \omega_k d(\mu_k, \lambda_k). \tag{12}$$

*Proof.* The proof is exactly as in Theorem 1 in Degenne & Koolen (2019). Specifically, in that paper, the result was stated with the expression of $T^\star(\boldsymbol{\mu})^{-1}$ given in Equation (11). Equation (12) follows by noticing that, for all $i \notin i^\star(\boldsymbol{\mu})$, then $\boldsymbol{\mu} \in \neg i$, and, hence, $\inf_{\boldsymbol{\lambda} \in \neg i} \sum_{k \in [K]} \omega_k d(\mu_k, \lambda_k) = 0$. □

## C    ON THE ASSUMPTIONS

In this section, we further discuss our assumptions. As mentioned in the main text, Assumption 1 is a mild requirement that is only needed for concentration arguments. Thus, in the following, we focus on Assumption 2. As our proofs show, Assumption 2 is only needed to bound differences of infimum of optimization problems which involve KL divergences. Specifically, it is employed only to control differences in KL for functions of the form:

$$\sum_k \omega_k (d(\mu_k, \lambda_{\mu,k}) - d(\mu'_k, \lambda_{\mu',k})), \tag{13}$$

where $\boldsymbol{\lambda}'_\mu$ is the minimizer of $\inf_{\boldsymbol{\lambda} \in \neg i} \omega_k d(\mu'_k, \lambda_k)$ for some $\neg i$. Therefore, our results holds for any family $\mathcal{M}$ of bandits for which it is possible to upper bound (in a Lipschitz fashion w.r.t. $\boldsymbol{\mu}$) functions of the form of Equation (13).

At this point, we remark on the following aspects.

- Degenne et al. (2019) originally provided the aforementioned intuitive relaxation of Assumption 2 and we invite the interested reader to check their Appendix F for further details.

- Degenne et al. (2019) also shows that for Gaussian setting on unbounded domains, Equation (13) can actually be bounded in a Lipschitz fashion. Hence, when dealing with Gaussian distributions, we can actually operate on unbounded domains (*i.e.*, we can remove Assumption 2).

Finally, we conclude by noticing that there exist works that have provided finite-confidence guarantees outside of Assumption 2. In particular:

- Jourdan & Degenne (2023) derived finite-confidence results for an optimistic variant of the Top-Two Algorithm without using Assumption 2. Nonetheless, the authors are restricting their analysis to Gaussian distributions, and, as we discussed above, our analysis can easily be generalized to cover this scenario.

- Barrier et al. (2022) also provides finite-confidence analysis outside of Assumption 2. Nonetheless, their non-asymptotic bounds feature an extra factor

$$\frac{1}{\omega_{\min}(\mu)^2} \exp(-\omega_{\min}(\mu)),$$

where $\omega_{min}(\mu) = \min_{k \in [K]} \omega_k^\star(\mu)$. We note that $\omega_{\min}$ can be lower-bounded using the minimum gap for Gaussian distributions (see the comment below Theorem 5 in Barrier et al. (2022)), and thus it is not an issue for Gaussian best-arm identification problems, as it can become arbitrary large only for instances which for which the sample complexity lower bound as well tends to $\infty$. Nonetheless, this is not the case for Bernoulli bandits. Indeed, consider a best-arm identification problem in a Bernoulli bandit scenario over 3 arms. Suppose that $\mu = (x, 0.8, 0.9)$. It is easy to see that $\omega_1 \to 0$ as $x \to 0$. Therefore, without Assumption 2, that finite-confidence guarantees can become vacuous outside of the Gaussian setting.

- Finally, Wang et al. (2021) also provided finite-confidence guarantees outside of Assumption 2; nonetheless, additional assumptions are needed in order to obtain the results. We refer the interested reader to Assumption 1-3 in Wang et al. (2021) for the technical requirements. Here, we only note that their finite-confidence analysis depend on assumptions that involve the gradients of the lower bound as a function of $\boldsymbol{\omega}$. Importantly, the main purpose of their assumptions is the same as ours, i.e., bounding functions of the form of (13). This is evident from Lemma 14 in Wang et al. (2021).

## D  Non-Asymptotic Bound for Track-and-Stop

In this section, we analyze the version of TAS that makes use of projection within the sampling rule. Specifically, $\boldsymbol{\omega}(s) \in \boldsymbol{\omega}^\star(\tilde{\boldsymbol{\mu}}(t))$, where $\tilde{\boldsymbol{\mu}}(t)$ denotes the orthogonal projection of $\hat{\boldsymbol{\mu}}(t)$ onto $[\mu_{\min}, \mu_{\max}]^K$. Before delving into the analysis, we note that it holds, due to the convexity of $d(\cdot, \cdot)$ (see, $e.g.$, Cappé et al. (2013)), we have that:

$$d(\hat{\mu}_k(t), \lambda) \geq d(\tilde{\mu}_k(t), \lambda), \quad \forall k \in [K], \lambda \in [\mu_{\min}, \mu_{\max}] \tag{14}$$

$$d(\lambda, \hat{\mu}_k(t)) \geq d(\lambda, \tilde{\mu}_k(t)) \quad \forall k \in [K], \lambda \in [\mu_{\min}, \mu_{\max}] \tag{15}$$

Now, we start by upper bounding the expectation of $\tau_\delta$ using an arbitrary good-event which implies stopping. The following result is standard in pure exploration works (see, $e.g.,$, Degenne et al. (2019)) and the proof is reported for completeness.

**Lemma 1** (Expectation Upper Bound). *Consider a sequence of events $\{\mathcal{E}_t\}_{t \geq 3}$ such that, there exists $T_0(\delta)$ and for $t \geq T_0(\delta)$ it holds that $\mathcal{E}_t \subseteq \{\tau_\delta \leq t\}$. Then, it holds that:*

$$\mathbb{E}_{\boldsymbol{\mu}}[\tau_\delta] \leq T_0(\delta) + \sum_{t=3}^{+\infty} \mathbb{P}_{\boldsymbol{\mu}}(\mathcal{E}_t^c).$$

*Proof.* It holds that:

$$\mathbb{E}_{\boldsymbol{\mu}}[\tau_\delta] = \sum_{t=0}^{+\infty} \mathbb{P}_{\boldsymbol{\mu}}(\tau_\delta > t) \leq 10K^4 + T_0(\delta) + \sum_{t=3+T_0(\delta)} \mathbb{P}_{\boldsymbol{\mu}}(\tau_\delta > t) \leq T_0(\delta) + \sum_{t=1}^{+\infty} \mathbb{P}_{\boldsymbol{\mu}}(\mathcal{E}_t^c).$$

$\square$

In our analysis, we will make use of the following good event:

$$\mathcal{E}_t = \left\{ \forall s \in \left[ \lceil \sqrt{t} \rceil, t \right], \sum_{k \in [K]} N_k(s) d(\hat{\mu}_k(s), \mu_k) \leq 8K \log(s) \right\}$$

Indeed, it can be shown with probabilistic arguments that $\sum_{t=3}^{+\infty} \mathbb{P}_{\boldsymbol{\mu}}(\mathcal{E}_t^c) \leq 2eK$ (Lemma 9). In the following, we compact the notation and we define $f(t) := 8K \log(t)$. The function $f(t)$ can be understood as an exploration function.

Next, the following lemma is the key result behind our analysis.

**Lemma 2** (Learning the Equilibrium (TAS)). *Let $t \geq 10K^4$. If TAS has not stopped at $t$, on $\mathcal{E}_t$, it holds that:*

$$\frac{\beta_{t,\delta}}{t} \geq \frac{t - \sqrt{t} - 1}{t} T^\star(\boldsymbol{\mu})^{-1} - \sum_{i=1}^{4} h_i(t)$$

*where:*

$$h_1(t) \leq \frac{D\sqrt{2\sigma^2 K f(t) t}}{t}$$

$$h_2(t) \leq \frac{LK^2 \ln(K)\sqrt{t + K^2}}{t}$$

$$h_3(t) \leq \frac{D\sqrt{2\sigma^2 f(t)}}{t}(K \ln K + 4\sqrt{Kt} + K^2\sqrt{t + K^2})$$

$$h_4(t) \leq \frac{D\sqrt{2\sigma^2 f(t)}}{t}\sqrt{8t^{3/2} + 8Kt \ln(t)}.$$

*Proof.* Let us define $h_1(t)$ as follows.

$$h_1(t) := \frac{1}{t} \left( \inf_{\boldsymbol{\lambda} \in \neg i^\star(\boldsymbol{\mu})} \sum_{k \in [K]} N_k(t) d(\mu_k, \lambda_k) - \inf_{\boldsymbol{\lambda} \in \neg i^\star(\boldsymbol{\mu})} \sum_{k \in [K]} N_k(t) d(\hat{\mu}_k(t), \lambda_k) \right)$$

If TAS has not stopped at $t \in \mathbb{N}$, then we have that:

$$\frac{\beta_{t,\delta}}{t} \geq \frac{1}{t} \max_{i \in \mathcal{I}} \inf_{\boldsymbol{\lambda} \in \neg i} \sum_{k \in [K]} N_k(t) d(\hat{\mu}_k(t), \lambda_k) \qquad \text{(Stopping Rule)}$$

$$\geq \frac{1}{t} \inf_{\boldsymbol{\lambda} \in \neg i^\star(\boldsymbol{\mu})} \sum_{k \in [K]} N_k(t) d(\hat{\mu}_k(t), \lambda_k)$$

$$\geq \frac{1}{t} \inf_{\boldsymbol{\lambda} \in \neg i^\star(\boldsymbol{\mu})} \sum_{k \in [K]} N_k(t) d(\mu_k, \lambda_k) - h_1(t). \qquad \text{(Definition of } h_1(t))$$

Next, we upper bound $h_1(t)$ on the good event $\mathcal{E}_t$.

$$h_1(t) = \frac{1}{t} \left( \inf_{\boldsymbol{\lambda} \in \neg i^\star(\boldsymbol{\mu})} \sum_{k \in [K]} N_k(t) d(\mu_k, \lambda_k) - \inf_{\boldsymbol{\lambda} \in \neg i^\star(\boldsymbol{\mu})} \sum_{k \in [K]} N_k(t) d(\hat{\mu}_k(t), \lambda_k) \right)$$

$$\leq \frac{1}{t} \sum_{k \in [K]} N_k(t) \sup_{\lambda \in \mathcal{M}} (d(\mu_k, \lambda) - d(\hat{\mu}_k(t), \lambda))$$

$$\leq \frac{1}{t} \sum_{k \in [K]} N_k(t) \sup_{\lambda \in \mathcal{M}} (\nu_{\mu_k} - \nu_\lambda) |\mu_k - \hat{\mu}_k(t)| \qquad \text{(Lemma 8)}$$

$$\leq \frac{D}{t} \sum_{k \in [K]} N_k(t) |\mu_k - \hat{\mu}_k(t)| \qquad \text{(Assumption 2)}$$

$$\leq \frac{D}{t} \sum_{k \in [K]} N_k(t) \sqrt{2\sigma^2 d(\hat{\mu}_k(t), \mu_k)} \qquad \text{(Assumption 1)}$$

$$\leq \frac{D}{t} \sum_{k \in [K]} N_k(t) \sqrt{2\sigma^2 \frac{f(t)}{N_k(t)}} \qquad \text{(On } \mathcal{E}_t, \text{ Lemma 9)}$$

$$\leq \frac{D\sqrt{2\sigma^2 K f(t) t}}{t}. \qquad \text{(By concavity of } \sqrt{\cdot})$$

We continue with a lower bound on $\frac{1}{t} \inf_{\boldsymbol{\lambda} \in \neg i^\star(\boldsymbol{\mu})} \sum_{k \in [K]} N_k(t) d(\mu_k, \lambda_k)$. Let $\{\boldsymbol{\omega}(s)\}_{s=1}^t$ be the sequence of empirical oracle weights selected by TAS, *i.e.*, $\boldsymbol{\omega}(s) \in \boldsymbol{\omega}^\star(\hat{\boldsymbol{\mu}}(t))$. Then, we have that:

$$\frac{1}{t} \inf_{\boldsymbol{\lambda} \in \neg i^\star(\boldsymbol{\mu})} \sum_{k \in [K]} N_k(t) d(\mu_k, \lambda_k) \geq \frac{1}{t} \inf_{\boldsymbol{\lambda} \in \neg i^\star(\boldsymbol{\mu})} \sum_{k \in [K]} \sum_{s=1}^t \omega_k(s) d(\mu_k, \lambda_k) - h_2(t),$$

where $h_2(t)$ is given by:

$$h_2(t) := \frac{1}{t} \inf_{\boldsymbol{\lambda} \in \neg i^\star(\boldsymbol{\mu})} \sum_{k \in [K]} \left( \sum_{s=1}^t \omega_k(s) - N_k(t) \right) d(\mu_k, \lambda_k)$$

$$\leq \frac{1}{t} K \ln(K) \sqrt{t + K^2} \inf_{\boldsymbol{\lambda} \in \neg i^\star(\boldsymbol{\mu})} \sum_{k \in [K]} d(\mu_k, \lambda_k) \qquad \text{(Lemma 7)}$$

$$:= \frac{L K^2 \ln(K) \sqrt{t + K^2}}{t}. \qquad \text{(Assumption 2)}$$

Then, we lower bound $\frac{1}{t} \inf_{\boldsymbol{\lambda} \in \neg i^\star(\boldsymbol{\mu})} \sum_{k \in [K]} \sum_{s=1}^t \omega_k(s) d(\mu_k, \lambda_k)$. To this end, we recall that, by definition:

$$\boldsymbol{\omega}(s) \in \operatorname*{argmax}_{\boldsymbol{\omega} \in \Delta_K} \max_{i \in \mathcal{I}} \inf_{\boldsymbol{\lambda} \in \neg i} \sum_{k \in [K]} \omega_k d(\tilde{\mu}_k(s), \lambda_k).$$

Let us denote by $i_s$ an answer that attains the argmax when paired with $\boldsymbol{\omega}(s)$. Then, we have that:

$$\frac{1}{t}\inf_{\boldsymbol{\lambda}\in\neg i^\star(\boldsymbol{\mu})}\sum_{k\in[K]}\sum_{s=1}^{t}\omega_k(s)d(\mu_k,\lambda_k) \geq \frac{1}{t}\sum_{s=1}^{t}\inf_{\boldsymbol{\lambda}\in\neg i^\star(\boldsymbol{\mu})}\sum_{k\in[K]}\omega_k(s)d(\mu_k,\lambda_k)$$

$$\geq \frac{1}{t}\sum_{s=1}^{t}\inf_{\boldsymbol{\lambda}\in\neg i_s}\sum_{k\in[K]}\omega_k(s)d(\mu_k,\lambda_k)$$

$$\geq \frac{1}{t}\sum_{s\geq\sqrt{t}}\inf_{\boldsymbol{\lambda}\in\neg i_s}\sum_{k\in[K]}\omega_k(s)d(\mu_k,\lambda_k)$$

$$\geq \frac{1}{t}\sum_{s\geq\sqrt{t}}\inf_{\boldsymbol{\lambda}\in\neg i_s}\sum_{k\in[K]}\omega_k(s)d(\tilde{\mu}_k(s),\lambda_k) - h_3(t),$$

where the second inequality follows from the fact that (i) if $i_s = i^\star(\boldsymbol{\mu})$ then the claim is trivial, and (ii) if $i_s \neq i^\star(\boldsymbol{\mu})$, then, $\boldsymbol{\mu} \in \neg i_s$ (since $i^\star(\boldsymbol{\mu})$ is single-valued) and $\inf_{\boldsymbol{\lambda}\in\neg i_s}\sum_{k\in[K]}\omega_k(s)d(\mu_k,\lambda_k) = 0$. Finally, the last step follows from the definition of $h_3(t)$, that is:

$$h_3(t) := \frac{1}{t}\sum_{s\geq\sqrt{t}}\inf_{\boldsymbol{\lambda}\in\neg i_s}\sum_{k\in[K]}\omega_k(s)d(\tilde{\mu}_k(s),\lambda_k) - \frac{1}{t}\sum_{s\geq\sqrt{t}}\inf_{\boldsymbol{\lambda}\in\neg i_s}\sum_{k\in[K]}\omega_k(s)d(\mu_k,\lambda_k)$$

$$\leq \frac{1}{t}\sum_{s\geq\sqrt{t}}\sum_{k\in[K]}\omega_k(s)\sup_{\boldsymbol{\lambda}\in\mathcal{M}}(d(\tilde{\mu}_k(s),\lambda_k) - d(\mu_k,\lambda_k))$$

$$\leq \frac{1}{t}\sum_{s\geq\sqrt{t}}\sum_{k\in[K]}\omega_k(s)\sup_{\boldsymbol{\lambda}\in\mathcal{M}}(\nu_{\tilde{\mu}_k(s)} - \nu_{\lambda_k})|\tilde{\mu}_k(s) - \mu_k| \qquad \text{(Lemma 8)}$$

$$\leq \frac{D}{t}\sum_{s\geq\sqrt{t}}\sum_{k\in[K]}\omega_k(s)|\tilde{\mu}_k(s) - \mu_k| \qquad \text{(Assumption 2 and } \tilde{\boldsymbol{\mu}}(s)\in[\mu_{\min},\mu_{\max}])$$

$$\leq \frac{D}{t}\sum_{s\geq\sqrt{t}}\sum_{k\in[K]}\omega_k(s)\sqrt{2\sigma^2 d(\tilde{\mu}_k(s),\mu_k)} \qquad \text{(Assumption 1)}$$

$$\leq \frac{D}{t}\sum_{s\geq\sqrt{t}}\sum_{k\in[K]}\omega_k(s)\sqrt{2\sigma^2 d(\hat{\mu}_k(s),\mu_k)} \qquad \text{(Equation (14))}$$

$$\leq \frac{D\sqrt{2\sigma^2 f(t)}}{t}\sum_{s\geq\sqrt{t}}\sum_{k\in[K]}\omega_k(s)\sqrt{\frac{1}{N_k(s)}} \qquad \text{(Lemma 9)}$$

$$\leq \frac{D\sqrt{2\sigma^2 f(t)}}{t}\left(K\ln(K) + 4\sqrt{Kt} + K^2\sqrt{t + K^2}\right). \qquad \text{(Lemma 7)}$$

We now have to analyze $\frac{1}{t}\sum_{s\geq\sqrt{t}}\inf_{\boldsymbol{\lambda}\in\neg i_s}\sum_{k\in[K]}\omega_k(s)d(\tilde{\mu}_k(s),\lambda_k)$.

Specifically, we have that:

$$\frac{1}{t}\sum_{s\geq\sqrt{t}}\inf_{\boldsymbol{\lambda}\in\neg i_s}\sum_{k\in[K]}\omega_k(s)d(\tilde{\mu}_k(s),\lambda_k) = \frac{1}{t}\sum_{s\geq\sqrt{t}}\sup_{\boldsymbol{\omega}\in\Delta_K}\max_{j\in[M]}\inf_{\boldsymbol{\lambda}\in\neg j}\sum_{k\in[K]}\omega_k(s)d(\tilde{\mu}_k(s),\lambda_k)$$

$$\geq \frac{1}{t}\sum_{s\geq\sqrt{t}}\inf_{\boldsymbol{\lambda}\in\neg i^\star(\boldsymbol{\mu})}\sum_{k\in[K]}\omega_k^\star d(\tilde{\mu}_k(s),\lambda_k)$$

$$\geq \frac{1}{t}\sum_{s\geq\sqrt{t}}\inf_{\boldsymbol{\lambda}\in\neg i^\star(\boldsymbol{\mu})}\sum_{k\in[K]}\omega_k^\star d(\mu_k,\lambda_k) - h_4(t)$$

$$= \frac{t-\sqrt{t}-1}{t}T^\star(\boldsymbol{\mu})^{-1} - h_4(t), \qquad \text{(Def. of } T^\star(\boldsymbol{\mu})^{-1})$$

where the first step follows from the definition of the sampling rule of TAS, in the second one we have chosen any $\omega^\star \in \omega^\star(\mu)$, and the third one by definition of $h_4(t)$, that is:

$$h_4(t) := \frac{1}{t} \sum_{s \geq \sqrt{t}} \inf_{\lambda \in \neg i^\star(\mu)} \sum_{k \in [K]} \omega_k^\star d(\mu_k, \lambda_k) - \frac{1}{t} \sum_{s \geq \sqrt{t}} \inf_{\lambda \in \neg i^\star(\mu)} \sum_{k \in [K]} \omega_k^\star d(\tilde{\mu}_k(s), \lambda_k)$$

Now, we conclude the proof by giving an upper bound on $h_4(t)$.

$$h_4(t) \leq \frac{1}{t} \sum_{s \geq \sqrt{t}} \sum_{k \in K} \omega_k^\star \sup_{\lambda \in \mathcal{M}} \left( d(\mu_k, \lambda_k) - d(\tilde{\mu}_k(s), \lambda_k) \right)$$

$$\leq \frac{1}{t} \sum_{s \geq \sqrt{t}} \sum_{k \in K} \omega_k^\star \sup_{\lambda \in \mathcal{M}} (\nu_{\mu_k} - \nu_{\lambda_k}) |\mu_k - \tilde{\mu}_k(s)| \qquad \text{(Lemma 8)}$$

$$\leq \frac{D}{t} \sum_{s \geq \sqrt{t}} \sum_{k \in K} \omega_k^\star |\mu_k - \tilde{\mu}_k(s)| \qquad \text{(Assumption 2)}$$

$$\leq \frac{D}{t} \sum_{s \geq \sqrt{t}} \| \mu - \tilde{\mu}(s) \|_\infty$$

$$\leq \frac{D}{t} \sum_{s \geq \sqrt{t}} \max_{k \in [K]} \sqrt{2\sigma^2 d(\tilde{\mu}_k(s), \mu_k)} \qquad \text{(Assumption 1)}$$

$$\leq \frac{D\sqrt{2\sigma^2 f(t)}}{t} \sum_{s \geq \sqrt{t}} \max_{k \in [K]} \sqrt{\frac{1}{N_k(s)}} \qquad \text{(Lemma 9)}$$

$$\leq \frac{D\sqrt{2\sigma^2 f(t)}}{t} \sum_{s \geq \sqrt{t}} \sqrt{\frac{1}{\sqrt{s + K^2} - 2K}} \qquad \text{(Lemma 7 and } t \geq 10K^4\text{)}$$

$$\leq \frac{D\sqrt{2\sigma^2 f(t)}}{t} \sqrt{t \sum_{s \geq \sqrt{t}} \frac{1}{\sqrt{s + K^2} - 2K}} \qquad \text{(Concavity of } \sqrt{\cdot} \text{ and } t \geq 10K^4\text{)}$$

$$\leq \frac{D\sqrt{2\sigma^2 f(t)}}{t} \sqrt{8t^{3/2} + 8Kt \ln(t)} \qquad \text{(Integral test and algebraic manipulations)}$$

which concludes the proof.[16] $\qquad\qquad \square$

*Proof of Theorem 1.* Let $t \geq 10K^4$. Then, for $t \geq 10K^4 + T_0(\delta)$, by Lemma 2, we have that $\mathcal{E}_t \subseteq \{\tau_\delta \leq t\}$. Lemma 1 and Lemma 9, then conclude the proof. To this end, it is sufficient to note that $T^\star(\mu) \sum_{i=1}^4 h(t)t + \sqrt{t} + 1 \leq g(t)$.[17] Indeed, using $t \geq 10^4$ and simple algebraic arguments, we have that:

$$th_1(t) \leq 4\sigma D\sqrt{Kt\log(t)} \leq 4\sigma DLK^2\log(K)\sqrt{t\log^2(t)}$$

$$th_2(t) \leq LK^2\log(K)\sqrt{2t} \leq DLK^2\log(K)\sqrt{2t\log^2(t)}$$

$$th_3(t) \leq 4\sigma D\sqrt{\log^2(t)}\left(K\log K + 4\sqrt{Kt} + K^2\sqrt{2t}\right) \leq 4\sigma DL\sqrt{\log^2(t)}\left(10K^2\log(K)\sqrt{t}\right)$$

$$th_4(t) \leq 16\sigma D\sqrt{Kt^{3/2}\log(t)} + 4\sigma DLK^2\log(K)\sqrt{8t\log^2(t)}.$$

Combining these inequalities, we obtain:

$$\sum_{i=1}^t th_i(t) \leq 62\sigma DLK^2\log(K)\sqrt{t\log^2(t)} + 16\sigma D\sqrt{Kt^{3/2}\log(t)}.$$

---

[16]The requirement of $t \geq 10K^4$ is essentially needed to guarantee that the denominators in those steps are always positive.

[17]In the relevant regime where $D, L, \sigma, T^\star(\mu) \geq 1$.

Thus, since from Lemma 2 we know that, for $t$ such that:

$$T^\star(\boldsymbol{\mu})\beta_{t,\delta} + \sqrt{t} + 1 + T^\star(\boldsymbol{\mu})\sum_{i=1}^{4} th_i(t) \leq t,$$

implies stopping on the good event, we also have that, TAS is guaranteed to stop whenever:

$$T^\star(\boldsymbol{\mu})\beta_{t,\delta} + T^\star(\boldsymbol{\mu})g(t) \leq t.$$

Rearranging the terms give the desired expression of $T_0(\delta)$. $\qquad\square$

## E   A SIMPLE FIX WITHOUT USING PROJECTIONS

In this section, we discuss what happens when TAS is not using projections in the sampling rule. The key idea is that there exists a time $T_\mathcal{M}$ such that, for subsequent steps $t$, then the empirical mean always lies within the interval $[\mu_{\min}, \mu_{\max}]$. Before that, we make a remark on Assumption 2. Specifically, fix any $\boldsymbol{\mu} \in \mathcal{M}$ and let $F_k := \min\{|\mu_k - \mu_{\min}|, |\mu_k - \mu_{\max}|\}$ and $F = \min_{k \in [K]} F_k$. As we discussed in Section 3, $F > 0$ holds to the fact that $\Theta$ is an open interval and $[\mu_{\min}, \mu_{\max}]$ is closed and contained in $\Theta$.

The following lemma shows the existence of such a $T_\mathcal{M}$.

**Lemma 3** (Empirical Means Lies in a Good Region). *Under Assumption 1 and Assumption 2, there exists a time $T_\mathcal{M} \in \mathbb{N}$ such that, for all $t \geq T_\mathcal{M}$, on $\mathcal{E}_t$, it holds that $\hat{\boldsymbol{\mu}}(s) \in [\mu_{\min}, \mu_{\max}]$ for all $s \geq \sqrt{t}$. Specifically,*

$$T_\mathcal{M} = \max\left\{10K^4, \inf\left\{n \in \mathbb{N} : \sqrt{\frac{64K\sigma^2\log(n)}{\sqrt{\sqrt{n}+K^2}-2K}} \leq F\right\}\right\}$$

*Proof.* Let $\bar{T}$ be such that, for all $t \geq \bar{T}$, $\sqrt{\sqrt{t}+K^2} - 2K$, *i.e.*, $\bar{T} \geq 10K^4$. Then, let $t \geq \bar{T}$.

Let $F_k := \min\{|\mu_k - \mu_{\min}|, |\mu_k - \mu_{\max}|\}$ and $F = \min_{k \in [K]} F_k$. Then, we have that if $\|\hat{\boldsymbol{\mu}}(t) - \boldsymbol{\mu}\|_\infty \leq F$, it holds that $\hat{\boldsymbol{\mu}}(t) \in [\mu_{\min}, \mu_{\max}]$. As discussed above, from Assumption 2, $F > 0$.

Now, on $\mathcal{E}_t$, for any $s \geq \sqrt{t}$, we have that:

$$\|\boldsymbol{\mu} - \hat{\boldsymbol{\mu}}(s)\|_\infty \leq \max_{k \in [K]} \sqrt{2\sigma^2 d(\hat{\mu}_k(s), \mu_k)} \qquad\qquad \text{(Assumption 1)}$$

$$\leq \max_{k \in [K]} \sqrt{\frac{2\sigma^2 f(s)}{N_k(s)}} \qquad\qquad \text{(Lemma 9)}$$

$$\leq \sqrt{\frac{2\sigma^2 f(s)}{\sqrt{s+K^2}-2K}} \qquad\qquad \text{(Lemma 7)}$$

$$\leq \sqrt{\frac{4\sigma^2 f(t)}{\sqrt{\sqrt{t}+K^2}-2K}}. \qquad\qquad (s \geq \sqrt{t} \text{ and } t \geq \bar{T})$$

Then, letting $T_\mathcal{M} = \max\left\{\bar{T}, \inf\left\{n \in \mathbb{N} : \sqrt{\frac{4\sigma^2 f(n)}{\sqrt{\sqrt{n}+K^2}-2K}} \leq F\right\}\right\}$ concludes the proof. $\quad\square$

Then, one can exploit Lemma 3 to obtain a result that is analogous to one of Theorem 1, just adding $T_\mathcal{M}$ to the finite-confidence upper bound on $\mathbb{E}_{\boldsymbol{\mu}}[\tau_\delta]$. Indeed, Lemma 2 holds as-is by analyzing any $t$ such that $t \geq T_\mathcal{M}$.

## F   NON-ASYMPTOTIC BOUND FOR STICKY TRACK-AND-STOP

In this section, we derive finite-confidence bounds for Sticky Track-and-Stop. We start with the following result, which shows the existence of a finite time after which (under the good event) the answer $i_t$ chosen by S-TAS follows within a "good set", *i.e.*, $i_F(\boldsymbol{\mu}) \cup (\mathcal{I} \setminus i^\star(\boldsymbol{\mu}))$.

**Lemma 4** (Good Answers on the Good Event). *Let $T_{\boldsymbol{\mu}}$ be defined as follows*

$$T = \max\left\{10K^4, \inf\left\{n \in \mathbb{N}, \sqrt{\frac{64K\sigma^2 \log(n)}{\sqrt{\sqrt{n} + K^2} - 2K}} \leq \epsilon_{\boldsymbol{\mu}}\right\}\right\},$$

*where $\epsilon_{\boldsymbol{\mu}} > 0$ is a problem dependent constant. Then, for all $t \geq T$, on $\mathcal{E}_t$, it holds that $i_s \in i_F(\boldsymbol{\mu}) \cup (\mathcal{I} \setminus i^\star(\boldsymbol{\mu}))$ for all $s \geq \sqrt{t}$.*

*Proof.* We recall that $\boldsymbol{\mu} \mapsto i_F(\boldsymbol{\mu})$ is upper hemicontinuous (Theorem 4 in Degenne & Koolen (2019)). This implies that there exists $\epsilon_{\boldsymbol{\mu}} > 0$ such that, for all $\boldsymbol{\mu}' : \|\boldsymbol{\mu} - \boldsymbol{\mu}'\|_\infty \leq \epsilon_{\boldsymbol{\mu}}$, it holds that $i_F(\boldsymbol{\mu}') \subseteq i_F(\boldsymbol{\mu}) \cup (\mathcal{I} \setminus i^\star(\boldsymbol{\mu}))$.

Now, consider $\bar{T}$ defined as follows:

$$\bar{T} = \inf\left\{n \in \mathbb{N} : \sqrt{\sqrt{n} + K^2} - 2K > 0\right\},$$

that is $\bar{T} = 10K^4$. Then, for all $t \geq \bar{T}$ and all $s \geq \sqrt{t}$, it holds that $\sqrt{s + K^2} - 2K > 0$.

Consider $t \geq \bar{T}$, and let us introduce, for all $\boldsymbol{\mu}, \boldsymbol{\mu}' \in \mathcal{M}$, $\mathrm{ch}(\boldsymbol{\mu}, \boldsymbol{\mu}') = \inf_{\boldsymbol{\lambda} \in \mathbb{R}^K} \sum_{k \in [K]} (d(\lambda_k, \mu_k) + d(\lambda_k, \mu'_k))$. Now, on $\mathcal{E}_t$ and for $s \geq \sqrt{t}$, we have that:

$$\sum_{k \in [K]} N_k(s) d(\hat{\mu}_k(s), \mu_k)) \leq 8K \log(s).$$

Furthermore, by definition, for all $\boldsymbol{\mu}' \in C_s$, we also have that:

$$\sum_{k \in [K]} N_k(s) d(\hat{\mu}_k(s), \mu'_k)) \leq 8K \log(s).$$

As a consequence, by applying Lemma 7, it holds that:

$$\mathrm{ch}(\boldsymbol{\mu}, \boldsymbol{\mu}') \left(\sqrt{s + K^2} - 2K\right) \leq \sum_{k \in} N_k(s) \left(d(\hat{\mu}_k(s), \mu_k) + d(\hat{\mu}_k(s), \mu'_k)\right) \leq 16K \log(s).$$

For $t \geq \bar{T}$, and using the definition of ch, this leads to:[18]

$$\frac{\|\boldsymbol{\mu} - \boldsymbol{\mu}'\|_\infty^2}{8\sigma^2} \leq \mathrm{ch}(\boldsymbol{\mu}, \boldsymbol{\mu}') \leq \frac{16K \log(s)}{\sqrt{s + K^2} - 2K}.$$

which leads to:

$$\|\boldsymbol{\mu} - \boldsymbol{\mu}'\|_\infty \leq \sqrt{\frac{32K\sigma^2 \log(s)}{\sqrt{s + K^2} - 2K}}, \quad \text{on } \mathcal{E}_t \ \forall s \geq \sqrt{t}, \boldsymbol{\mu}' \in C_s.$$

Thus, for $t \geq \max\left\{\bar{T}, \inf\left\{n \in \mathbb{N}, \sqrt{\frac{64K\sigma^2 \log(n)}{\sqrt{\sqrt{n} + K^2} - 2K}} \leq \epsilon_{\boldsymbol{\mu}}\right\}\right\}$, it holds that:

$$\|\boldsymbol{\mu} - \boldsymbol{\mu}'\|_\infty \leq \epsilon_{\boldsymbol{\mu}}, \quad \text{on } \mathcal{E}_t \ \forall s \geq \sqrt{t}, \boldsymbol{\mu}' \in C_s.$$

Now, since $i_s \in \mathcal{I}_s = \bigcup_{\boldsymbol{\mu}' \in C_s} i_F(\boldsymbol{\mu}')$ and $\|\boldsymbol{\mu} - \boldsymbol{\mu}'\|_\infty \leq \epsilon_{\boldsymbol{\mu}}$ for all $\boldsymbol{\mu}' \in C_s$, it follows (by definition of $\epsilon_{\boldsymbol{\mu}}$) that, on $\mathcal{E}_t$, for $s \geq \sqrt{t}$, $i_s \in i_F(\boldsymbol{\mu}) \cup (\mathcal{I} \setminus i^\star(\boldsymbol{\mu}))$, thus concluding the proof. □

Next, the following result is the key lemma that provides a lower bound, under the good event $\mathcal{E}_t$, on the information gathered by S-TAS.

---

[18]The lower bound on $\mathrm{ch}(\cdot, \cdot)$ follows from using the sub-gaussianity of the arms to lower bound the divergences $d$ with the difference in means and solving the resulting $\inf$ problem over $\mathbb{R}^K$.

**Lemma 5** (Learning the Equilibrium (S-TAS)). *Let $t \geq 10K^4$ and let $T_{\boldsymbol{\mu}}$ as in Lemma 4. Define $\widetilde{T} = \max\{T_{\boldsymbol{\mu}}, \lceil \sqrt{t} \rceil\}$. Then, for S-TAS, on $\mathcal{E}_t$, it holds that:*

$$\frac{\beta_{t,\delta}}{t} \geq \frac{t - \widetilde{T}}{t} T^\star(\boldsymbol{\mu})^{-1} - \sum_{i=1}^{5} h_i(t),$$

*where*

$$h_1(t) \leq \frac{D\sqrt{2\sigma^2 K f(t) t}}{t}$$

$$h_2(t) \leq \frac{LK^2 \ln(K)\sqrt{t + K^2}}{t}$$

$$h_3(t) \leq \frac{D\sqrt{2\sigma^2 f(t)}}{t}\left( K\ln(K) + 4\sqrt{Kt} + K^2\sqrt{t + K^2} \right)$$

$$h_4(t) \leq \frac{D\sqrt{2\sigma^2 f(t)}}{t}\sqrt{8t^{3/2} + 8Kt\ln(t)}$$

$$h_5(t) \leq \frac{2D\sqrt{2\sigma^2 f(t)}}{t}\sqrt{8t^{3/2} + 8Kt\ln(t)}.$$

*Proof.* Let $h_1(t)$ be defined as follows:

$$h_1(t) = \frac{1}{t} \max_{i \in \mathcal{I}} \inf_{\boldsymbol{\lambda} \in \neg i} \sum_{k \in [K]} N_k(t) d(\mu_k, \lambda_k) - \frac{1}{t} \max_{i \in \mathcal{I}} \inf_{\boldsymbol{\lambda} \in \neg i} \sum_{k \in [K]} N_k(t) d(\hat{\mu}_k(t), \lambda_k)$$

If Sticky Track-and-Stop has not stopped at $t \in \mathbb{N}$, then, it holds that:

$$\frac{\beta_{t,\delta}}{t} \geq \frac{1}{t} \max_{i \in \mathcal{I}} \inf_{\boldsymbol{\lambda} \in \neg i} \sum_{k \in [K]} N_k(t) d(\hat{\mu}_k(t), \lambda_k) \qquad \text{(Stopping Rule)}$$

$$= \frac{1}{t} \max_{i \in \mathcal{I}} \inf_{\boldsymbol{\lambda} \in \neg i} \sum_{k \in [K]} N_k(t) d(\mu_k(t), \lambda_k) - h_1(t). \qquad \text{(Definition of $h_1(t)$)}$$

Next, we upper bound $h_1(t)$ under the good event $\mathcal{E}_t$.

$$h_1(t) \leq \frac{1}{t} \sum_{k \in [K]} N_k(t) \sup_{\boldsymbol{\lambda} \in \mathcal{M}} \left( d(\mu_k, \lambda_k) - d(\hat{\mu}_k(t), \lambda_k) \right)$$

$$\leq \frac{1}{t} \sum_{k \in [K]} N_k(t) \sup_{\boldsymbol{\lambda} \in \mathcal{M}} \left( d(\mu_k, \lambda_k) - d(\hat{\mu}_k(t), \lambda_k) \right)$$

$$\leq \frac{1}{t} \sum_{k \in [K]} N_k(t) \sup_{\boldsymbol{\lambda} \in \mathcal{M}} (\nu_{\mu_k} - \nu_{\lambda_k})|\mu_k - \hat{\mu}_k(t)| \qquad \text{(Lemma 8)}$$

$$\leq \frac{D}{t} \sum_{k \in [K]} N_k(t)|\mu_k - \hat{\mu}_k(t)| \qquad \text{(Assumption 2)}$$

$$\leq \frac{D}{t} \sum_{k \in [K]} N_k(t)\sqrt{2\sigma^2 d(\hat{\mu}_k(t), \mu_k)} \qquad \text{(Assumption 1)}$$

$$\leq \frac{D}{t} \sum_{k \in [K]} N_k(t)\sqrt{2\sigma^2 \frac{f(t)}{N_k(t)}} \qquad \text{(Lemma 9)}$$

$$\leq \frac{D\sqrt{2\sigma^2 K f(t) t}}{t}. \qquad \text{(Concavity of $\sqrt{\cdot}$)}$$

Next, we continue by analyzing $\frac{1}{t}\max_{i\in\mathcal{I}}\inf_{\boldsymbol{\lambda}\in\neg i}\sum_{k\in[K]}N_k(t)d(\mu_k(t),\lambda_k)$. Let $\{\boldsymbol{\omega}(s)\}_s$ be the sequence of empirical oracle weights computed by Sticky Track-and-Stop. To this end, let $h_2(t)$ be defined as follows:

$$h_2(t) = \frac{1}{t}\max_{i\in\mathcal{I}}\inf_{\boldsymbol{\lambda}\in\neg i}\sum_{k\in[K]}\left(\sum_{s=1}^{t}\omega_k(s) - N_k(t)\right)d(\mu_k,\lambda_k).$$

Then, by definition of $h_2(t)$, we have that:

$$\frac{1}{t}\max_{i\in\mathcal{I}}\inf_{\boldsymbol{\lambda}\in\neg i}\sum_{k\in[K]}N_k(t)d(\mu_k(t),\lambda_k) \geq \frac{1}{t}\max_{i\in\mathcal{I}}\inf_{\boldsymbol{\lambda}\in\neg i}\sum_{s=1}^{t}\sum_{k\in[K]}\omega_k(s)d(\mu_k,\lambda_k) - h_2(t)$$

Next, we upper bound $h_2(t)$.

$$h_2(t) \leq \frac{1}{t}\max_{i\in\mathcal{I}}\inf_{\boldsymbol{\lambda}\in\neg i}\sum_{k\in[K]}K\ln(K)\sqrt{t+K^2}d(\mu_k,\lambda_k) \qquad \text{(Lemma 7)}$$

$$\leq \frac{LK^2\ln(K)\sqrt{t+K^2}}{t}. \qquad \text{(Assumption 2)}$$

Next, we focus on $\frac{1}{t}\max_{i\in\mathcal{I}}\inf_{\boldsymbol{\lambda}\in\neg i}\sum_{s=1}^{t}\sum_{k\in[K]}\omega_k(s)d(\mu_k,\lambda_k)$. Let $\imath\in\mathcal{I}$ be such that, given the subset of answers $i_F(\boldsymbol{\mu})$, then, the pre-specified total order over $\mathcal{I}$ selects $\imath$. Furthermore, let $T_{\boldsymbol{\mu}}$ be as in Lemma 4 and let $\widetilde{T} = \max\{T_{\boldsymbol{\mu}}, \lceil\sqrt{t}\rceil\}$ Then, it holds that:

$$\frac{1}{t}\max_{i\in\mathcal{I}}\inf_{\boldsymbol{\lambda}\in\neg i}\sum_{s=1}^{t}\sum_{k\in[K]}\omega_k(s)d(\mu_k,\lambda_k) \geq \frac{1}{t}\sum_{s=1}^{t}\inf_{\boldsymbol{\lambda}\in\neg\imath}\sum_{k\in[K]}\omega_k(s)d(\mu_k,\lambda_k)$$

$$\geq \frac{1}{t}\sum_{s=\widetilde{T}}^{t}\inf_{\boldsymbol{\lambda}\in\neg\imath}\sum_{k\in[K]}\omega_k(s)d(\mu_k,\lambda_k)$$

$$\geq \frac{1}{t}\sum_{s=\widetilde{T}}^{t}\inf_{\boldsymbol{\lambda}\in\neg i_s}\omega_k(s)d(\mu_k,\lambda_k),$$

where in the last step, we have used the fact that, (i) if $i_s\in i_F(\boldsymbol{\mu})$, then $i_s=\imath$ on the good event $\mathcal{E}_t$ [19] and (ii) if $i_s\notin i_F(\boldsymbol{\mu})$, then $i_s\notin i^{\star}(\boldsymbol{\mu})$ due to the definition of $\overline{T}$. Then, in this case, we have that $\boldsymbol{\mu}\in\neg i_s$ and $\inf_{\boldsymbol{\lambda}\in\neg i_s}\omega_k(s)d(\mu_k,\lambda_k) = 0$. Next, we have that:

$$\frac{1}{t}\sum_{s=\widetilde{T}}^{t}\inf_{\boldsymbol{\lambda}\in\neg i_s}\sum_{k\in[K]}\omega_k(s)d(\mu_k,\lambda_k) \geq \frac{1}{t}\sum_{s=\widetilde{T}}^{t}\inf_{\boldsymbol{\lambda}\in\neg i_s}\sum_{k\in[K]}\omega_k(s)d(\tilde{\mu}_k(s),\lambda_k) - h_3(t),$$

where $h_3(t)$ is given by:

$$h_3(t) := \frac{1}{t}\sum_{s=\widetilde{T}}^{t}\inf_{\boldsymbol{\lambda}\in\neg i_s}\sum_{k\in[K]}\omega_k(s)d(\tilde{\mu}_k(s),\lambda_k) - \frac{1}{t}\sum_{s=\widetilde{T}}^{t}\inf_{\boldsymbol{\lambda}\in\neg i_s}\sum_{k\in[K]}\omega_k(s)d(\mu_k,\lambda_k)$$

---

[19]Indeed, in that case, $\boldsymbol{\mu}\in C_s$, and, consequently, $i_F(\boldsymbol{\mu})\in\mathcal{I}_s$. Since the algorithm selects answers according to a total order, it cannot select any answer in $i_F(\boldsymbol{\mu})$ which is not $\imath$.

Now, we have that:

$$
\begin{aligned}
h_3(t) &\leq \frac{1}{t} \sum_{s=\widetilde{T}}^{t} \sum_{k \in [K]} \omega_k(s) \sup_{\boldsymbol{\lambda} \in \mathcal{M}} \left(d(\tilde{\mu}_k(s), \lambda_k) - d(\mu_k, \lambda_k)\right) \\
&\leq \frac{D}{t} \sum_{s=\widetilde{T}}^{} \sum_{k \in [K]} \omega_k(s) |\hat{\mu}_k(s) - \mu_k| && \text{(Lemma 8, Assumption 2, def. of } \tilde{\mu}_k(s)) \\
&\leq \frac{D}{t} \sum_{s=\widetilde{T}}^{t} \sum_{k \in [K]} \omega_k(s) \sqrt{2\sigma^2 d(\hat{\mu}_k(s), \mu_k)} && \text{(Assumption 1)} \\
&\leq \frac{D\sqrt{2\sigma^2 f(t)}}{t} \sum_{s=1}^{t} \sum_{k \in [K]} \omega_k(s) \sqrt{\frac{1}{N_k(s)}} && \text{(Lemma 9)} \\
&\leq \frac{D\sqrt{2\sigma^2 f(t)}}{t} \left(K \ln(K) + 4\sqrt{Kt} + K^2\sqrt{t + K^2}\right). && \text{(Lemma 7)}
\end{aligned}
$$

Now, we continue by analyzing

$$
\frac{1}{t} \sum_{s=\widetilde{T}}^{t} \inf_{\boldsymbol{\lambda} \in \neg i_s} \sum_{k \in [K]} \omega_k(s) d(\tilde{\mu}_k(s), \lambda_k).
$$

Specifically, by definition of $i_s$ and $\boldsymbol{\omega}(s)$, we have that:

$$
\begin{aligned}
\frac{1}{t} \sum_{s=\widetilde{T}}^{t} \inf_{\boldsymbol{\lambda} \in \neg i_s} \sum_{k \in [K]} \omega_k(s) d(\tilde{\mu}_k(s), \lambda_k) &= \frac{1}{t} \sum_{s=\widetilde{T}}^{t} \max_{\boldsymbol{\omega} \in \Delta_K} \inf_{\boldsymbol{\lambda} \in \neg i_s} \sum_{k \in [K]} \omega_k d(\tilde{\mu}_k(s), \lambda_k) \\
&= \frac{1}{t} \sum_{s=\widetilde{T}}^{t} \max_{\boldsymbol{\omega} \in \Delta_K} \inf_{\boldsymbol{\lambda} \in \neg i_s} \sum_{k \in [K]} \omega_k d(\mu'_k(s), \lambda_k) - h_4(t),
\end{aligned}
$$

where $\boldsymbol{\mu}'(s) \in \mathcal{M}$ is such that $i_s \in i_F(\boldsymbol{\mu}'(s))$ and $h_4(t)$ is given by:

$$
h_4(t) := \frac{1}{t} \sum_{s=\widetilde{T}}^{t} \max_{\boldsymbol{\omega} \in \Delta_K} \inf_{\boldsymbol{\lambda} \in \neg i_s} \sum_{k \in [K]} \omega_k d(\mu'_k(s), \lambda_k) - \frac{1}{t} \sum_{s=\widetilde{T}}^{t} \max_{\boldsymbol{\omega} \in \Delta_K} \inf_{\boldsymbol{\lambda} \in \neg i_s} \sum_{k \in [K]} \omega_k d(\tilde{\mu}_k(s), \lambda_k).
$$

Now, we upper bound $h_4(t)$.

$$
\begin{aligned}
h_4(t) &\le \frac{1}{t} \sum_{s=\widetilde{T}}^{t} \max_{\boldsymbol{\omega} \in \Delta_k} \left( \inf_{\boldsymbol{\lambda} \in \neg i_s} \sum_{k \in [K]} \omega_k d(\mu'_k(s), \lambda_k) - \inf_{\boldsymbol{\lambda} \in \neg i_s} \sum_{k \in [K]} \omega_k d(\tilde{\mu}_k(s), \lambda_k) \right) \\
&\le \frac{1}{t} \sum_{s=\widetilde{T}}^{t} \max_{\boldsymbol{\omega} \in \Delta_k} \left( \sum_{k \in [K]} \omega_k \sup_{\boldsymbol{\lambda} \in \mathcal{M}} \left( d(\mu'_k(s), \lambda_k) - d(\tilde{\mu}_k(s), \lambda_k) \right) \right) \\
&\le \frac{1}{t} \sum_{s=\widetilde{T}}^{t} \max_{\boldsymbol{\omega} \in \Delta_K} \sum_{k \in [K]} \omega_k \sup_{\boldsymbol{\lambda} \in \mathcal{M}} \left( \nu_{\mu'_k(s)} - \nu_{\lambda_k} \right) |\mu'_k(s) - \tilde{\mu}_k(s)| && \text{(Lemma 8)} \\
&\le \frac{D}{t} \sum_{s=\widetilde{T}}^{t} \max_{\boldsymbol{\omega} \in \Delta_K} \sum_{k \in [K]} \omega_k |\mu'_k(s) - \hat{\mu}_k(s)| && \text{(Assumption 2)} \\
&\le \frac{D}{t} \sum_{s=\widetilde{T}}^{t} \max_{\boldsymbol{\omega} \in \Delta_K} \sum_{k \in [K]} \omega_k \sqrt{2\sigma^2 d(\hat{\mu}_k(s), \mu'_k(s))} && \text{(Assumption 1)} \\
&\le \frac{D\sqrt{2\sigma^2 f(t)}}{t} \sum_{s=\widetilde{T}}^{t} \max_{\boldsymbol{\omega} \in \Delta_K} \sum_{k \in [K]} \omega_k \sqrt{\frac{1}{N_k(s)}} && (\mu'_k(s) \in C_s) \\
&\le \frac{D\sqrt{2\sigma^2 f(t)}}{t} \sum_{s=\widetilde{T}}^{t} \sqrt{\frac{1}{\sqrt{s+K^2-2K}}} && \text{(Lemma 7)} \\
&\le \frac{D\sqrt{2\sigma^2 f(t)}}{t} \sqrt{t \sum_{s=\widetilde{T}}^{t} \frac{1}{\sqrt{s+K^2-2K}}} && \text{(Concavity of } \sqrt{\cdot}) \\
&\le \frac{D\sqrt{2\sigma^2 f(t)}}{t} \sqrt{8t^{3/2} + 8Kt\ln(t)} && \text{(Integral test and algebraic manipulations)}
\end{aligned}
$$

Next, it remains to analyze $\frac{1}{t} \sum_{s=\widetilde{T}}^{t} \max_{\boldsymbol{\omega} \in \Delta_K} \inf_{\boldsymbol{\lambda} \in \neg i_s} \sum_{k \in [K]} \omega_k d(\mu'_k(s), \lambda_k)$. Let $\boldsymbol{\omega}^\star \in \boldsymbol{\omega}^\star(\boldsymbol{\mu}, \neg \imath)$. Then, we have that:

$$
\begin{aligned}
\frac{1}{t} \sum_{s=\widetilde{T}}^{t} \max_{\boldsymbol{\omega} \in \Delta_K} \inf_{\boldsymbol{\lambda} \in \neg i_s} \sum_{k \in [K]} \omega_k d(\mu'_k(s), \lambda_k) &= \frac{1}{t} \sum_{s=\widetilde{T}}^{t} \max_{i \in \mathcal{I}} \max_{\boldsymbol{\omega} \in \Delta_K} \inf_{\boldsymbol{\lambda} \in \neg i} \sum_{k \in [K]} \omega_k d(\mu'_k(s), \lambda_k) \\
&\ge \frac{1}{t} \sum_{s=\widetilde{T}}^{t} \inf_{\boldsymbol{\lambda} \in \neg \imath} \sum_{k \in [K]} \omega_k^\star d(\mu'_k(s), \lambda_k) \\
&\ge \frac{1}{t} \sum_{s=\widetilde{T}}^{t} \inf_{\boldsymbol{\lambda} \in \neg \imath} \sum_{k \in [K]} \omega_k^\star d(\mu_k, \lambda_k) - h_5(t) \\
&= \frac{t - \widetilde{T}}{t} T^\star(\boldsymbol{\mu})^{-1} - h_5(t),
\end{aligned}
$$

where the first step follows from the fact that $i_s \in i_F(\boldsymbol{\mu}'(s))$ and the last third one from the definition of $h_5(t)$, that is:

$$
\begin{aligned}
h_5(t) &:= \frac{1}{t} \sum_{s=\widetilde{T}}^{t} \inf_{\boldsymbol{\lambda} \in \neg i} \sum_{k \in [K]} \omega_k^\star d(\mu_k, \lambda_k) - \frac{1}{t} \sum_{s=\widetilde{T}}^{t} \inf_{\boldsymbol{\lambda} \in \neg i} \sum_{k \in [K]} \omega_k^\star d(\mu_k'(s), \lambda_k) \\
&\leq \frac{1}{t} \sum_{s=\widetilde{T}}^{t} \sum_{k \in [K]} \omega_k^\star \sup_{\boldsymbol{\lambda} \in \mathcal{M}} \left( d(\mu_k, \lambda_k) - d(\mu_k'(s), \lambda_k) \right) \\
&\leq \frac{1}{t} \sum_{s=\widetilde{T}}^{t} \sum_{k \in [K]} \omega_k^\star \sup_{\boldsymbol{\lambda} \in \mathcal{M}} (\nu_{\mu_k} - \nu_{\lambda_k}) |\mu_k - \mu_k'(s)| && \text{(Lemma 8)} \\
&\leq \frac{D}{t} \sum_{s=\widetilde{T}}^{t} \sum_{k \in [K]} \omega_k^\star |\mu_k - \mu_k'(s)| && \text{(Assumption 2)} \\
&\leq \frac{D}{t} \sum_{s=\widetilde{T}}^{t} \|\boldsymbol{\mu} - \boldsymbol{\mu}'(s)\|_\infty \\
&\leq \frac{D}{t} \sum_{s=\widetilde{T}}^{t} \|\boldsymbol{\mu} - \hat{\boldsymbol{\mu}}(s)\|_\infty + \|\hat{\boldsymbol{\mu}}(s) - \boldsymbol{\mu}'(s)\|_\infty \\
&\leq \frac{2D\sqrt{2\sigma^2 f(t)}}{t} \sum_{s=\widetilde{T}}^{t} \max_{k \in [K]} \frac{1}{\sqrt{N_k(s)}} && \text{(Lemma 9, Assumption 1, } \boldsymbol{\mu}'(s) \in C_s) \\
&\leq \frac{2D\sqrt{2\sigma^2 f(t)}}{t} \sum_{s=\widetilde{T}}^{t} \frac{1}{\sqrt{\sqrt{s + K^2} - 2K}} && \text{(Lemma 7)} \\
&\leq \frac{2D\sqrt{2\sigma^2 f(t)}}{t} \sqrt{t \sum_{s=\widetilde{T}}^{t} \frac{1}{\sqrt{s + K^2} - 2K}} && \text{(Concavity of } \sqrt{\cdot}) \\
&\leq \frac{2D\sqrt{2\sigma^2 f(t)}}{t} \sqrt{8t^{3/2} + 8Kt\ln(t)}, && \text{(Integral test and algebraic manipulations)}
\end{aligned}
$$

which concludes the proof. $\qquad\square$

We are now ready to prove Theorem 2.

*Proof of Theorem 2.* Let $t \geq K^2$. Then, for $t \geq 10K^4 + T_0(\delta)$, by Lemma 5, we have $\mathcal{E}_t \subseteq \{\tau_\delta \leq t\}$. Lemma 1 and Lemma 9 then conclude the proof. To this end, it is sufficient to note that $T^\star(\boldsymbol{\mu}) \sum_{i=1}^{5} h(t)t + \sqrt{t} + 1 \leq g(t)$.[20] Here, we followed the same algebraic steps that we presented for the proof of Theorem 1. $\qquad\square$

# G  AN EXPLICIT BOUND

One concern with the implicit definition of $T_0(\delta)$ in Theorems 1 and 2 is whether they yield practically useful rates. In this appendix, we demonstrate that they do. By carefully bounding $T_0(\delta)$, we show that it can be upper-bounded by clean and explicit expression of the form

$$
\mathcal{O}(T^\star(\boldsymbol{\mu}) \log(1/\delta) + T^\star(\boldsymbol{\mu}) K \log\log(1/\delta)).
$$

These bounds are nearly tight: they match the asymptotic lower bound up to polylogarithmic factors and additive problem-dependent constants. Furthermore, they are also useful in understanding the theoretical guarantees of the algorithms in the regime where $T^\star(\boldsymbol{\mu}) \to +\infty$ (*i.e.*, in a BAI problem,

---

[20]In the relevant regime where $D, L, \sigma, T^\star(\boldsymbol{\mu}) \geq 1$.

these are the hard instances where the mean of the second best arm approaches the one of the optimal arm).

This section is structured as follows. First, in Appendix G.1 we present the result of TAS, then in Appendix G.2 we present the result for S-TAS.

### G.1 AN EXPLICIT BOUND FOR TAS

For TAS, we show the following upper-bound on $T_0(\delta)$.

**Proposition 3.** *Consider any $\eta_1 \in (0, 1/2)$, $\eta_2 \in (0, 1/4)$ and let $A_1(\eta_1)$ and $A_2(\eta_2)$ be defined as follows:*

$$A_1(\eta_1) \coloneqq \frac{66\sigma DLK^2 \log(K)T^\star(\boldsymbol{\mu})}{\eta_1} \qquad A_2(\eta_2) \coloneqq \frac{16\sigma D\sqrt{K}T^\star(\boldsymbol{\mu})}{\eta_2}.$$

*Furthermore, consider any $\alpha, \gamma \in (0, 1)$ such that $\alpha + \gamma < 1$ and let $\tilde{A}_1(\eta, \alpha)$ and $\tilde{A}_2(\eta, \gamma)$ be defined as follows:*

$$\tilde{A}_1(\eta_1, \alpha) \coloneqq A_1(\eta_1) \left( \frac{(0.5 + \eta_1)A_1(\eta_1)}{\alpha} \right)^{\frac{0.5+\eta_1}{0.5-\eta_1}} \qquad \tilde{A}_2(\eta_2, \gamma) \coloneqq A_2(\eta_2) \left( \frac{(0.75 + \eta_2)A_2(\eta_2)}{\gamma} \right)^{\frac{0.75+\eta_2}{0.25-\eta_2}}$$

*Then, it holds that:*

$$T_0(\delta) \leq \frac{T^\star(\boldsymbol{\mu})\log(1/\delta) + T^\star(\boldsymbol{\mu})K\log(\log(1/\delta) + 1) + \tilde{A}_1(\eta_1, \alpha) + \tilde{A}_2(\eta_2, \gamma)}{1 - \alpha - \gamma}. \tag{16}$$

Before proving the proposition, we comment on the result.

Proposition 3 provides an upper bound on $T_0(\delta)$ that holds for all $\eta_1, \eta_2, \alpha, \gamma > 0$ such that $\eta_1 < 1/2$, $\eta_2 < 1/4$ and $\alpha + \gamma < 1$. Hence, the tightest bound is achieved while minimizing Equation (16) over this domain. Some comments are in order.

**Asymptotic Regime of $\delta \to 0$** First, whenever $\delta \to 0$, one can pick any valid $\eta_1, \eta_2$ together with $\alpha$ and $\gamma$ that goes progressively to 0, yielding to $T_0(\delta) \approx T^\star(\boldsymbol{\mu})\log(1/\delta) + T^\star(\boldsymbol{\mu})K\log(\log(1/\delta) + 1)$. This shows how our result retrieves asymptotic optimality together with the dependency on minor order terms of $\delta$.

**Asymptotic Regime of $T^\star(\boldsymbol{\mu}) \to +\infty$** Second, by picking $\alpha = \gamma = \frac{1}{4}$, we can easily evaluate the moderate regime of $\delta$ in difficult instances where $T^\star(\boldsymbol{\mu}) \to \infty$ (e.g., the case of best-arm identification where the gap between optimal arm and the second best one tends to 0). In this case, one obtains a rate of the form $\mathcal{O}\left( A_1(\eta_1)^{1 + \frac{0.5+\eta_1}{0.5-\eta_1}} + A_2(\eta_2)^{1 + \frac{0.75+\eta_2}{0.25-\eta_2}} \right)$. To further understand the dependencies in relevant quantities, let us first analyze the first term, that is:

$$\inf_{\eta \in (0, \frac{1}{2})} A_1(\eta)^{1 + \frac{0.5+\eta}{0.5-\eta}} = \inf_{\eta \in (0, \frac{1}{2})} A_1(\eta)^{\frac{1}{0.5-\eta}}$$

$$= \inf_{\eta \in (0, \frac{1}{2})} \left( \frac{66\sigma DLK^2 \log(K)T^\star(\boldsymbol{\mu})}{\eta} \right)^{\frac{1}{0.5-\eta}}$$

$$\coloneqq \inf_{\eta \in (0, \frac{1}{2})} \left( \frac{A_1}{\eta} \right)^{\frac{1}{0.5-\eta}},$$

where in the last step we introduced $A_1 := 66\sigma DLK^2 \log(K) T^\star(\boldsymbol{\mu})$. Let $L_1 = \log(4A_1)$. Then, for sufficiently large $T^\star(\boldsymbol{\mu})$, we have that $\frac{1}{4L_1} \in (0, \frac{1}{2})$. Hence, we obtain that:

$$
\begin{aligned}
\inf_{\eta \in (0, \frac{1}{2})} \left(\frac{A_1}{\eta}\right)^{\frac{1}{0.5-\eta}} &\leq (4A_1 L_1)^{\frac{1}{0.5 - \frac{1}{4L_1}}} \\
&= (4A_1 L_1)^{\frac{4L_1}{2L_1-1}} \\
&= (4A_1 L_1)^2 (4A_1 L_1)^{\frac{2}{2L_1-1}} \\
&= (4A_1 L_1)^2 \exp\left(\frac{2}{2L_1-1} \log(4A_1 L_1)\right) \\
&= (4A_1 L_1)^2 \exp\left(\frac{4L_1-2}{2L_1-1}\right) \\
&\in \mathcal{O}\left(A_1^2 \log(A_1)^2\right),
\end{aligned}
$$

where, in the forth step, we have used that for $T^\star(\boldsymbol{\mu})$ sufficiently large $4L_1 - 2 \geq 2L_1 + 2\log(L_1)$.

Following similar reasoning for the term with $A_2$, we have that:

$$
\inf_{\eta \in (0, \frac{1}{4})} A_2(\eta)^{1 + \frac{0.75+\eta}{0.25-\eta}} \leq \mathcal{O}\left(A_2^4 \log(A_2)^4\right),
$$

where $A_2 := 16\sigma D\sqrt{K} T^\star(\boldsymbol{\mu})$.

Hence, we have that:

$$
\inf_{\eta_1 \in (0, 1/2), \eta_2 \in (0, 1/4)} A_1(\eta_1)^{1 + \frac{0.5+\eta_1}{0.5-\eta_1}} + A_2(\eta_2)^{1 + \frac{0.75+\eta_2}{0.25-\eta_2}} \leq 2 \max\left\{A_1^2 \log(A_1)^2, A_2^4 \log(A_2)^4\right\}
$$

For $T^\star(\boldsymbol{\mu}) \to \infty$, $A_2^4 \log(A_2)^4$ is the dominant term, thus providing the relevant dependencies in term of $D, \sigma, L, K$ and $T^\star(\boldsymbol{\mu})$.

We now conclude this section with a proof of Proposition 3.

*Proof of Proposition 3.* From Theorem 1, we have that:

$$
T_0(\delta) = \inf\left\{t \in \mathbb{N} : \beta_{t,\delta} \leq (t - \sqrt{t} - 1)T^\star(\boldsymbol{\mu})^{-1} - g(t)\right\},
$$

where $g(t)$ is given by:

$$
g(t) = 64\sigma DLK^2 \log(K)\sqrt{t \log^2(t)} + 16D\sqrt{Kt^{3/2}\log(t)}.
$$

Using $\log(t) \leq \frac{t^\eta}{\eta}$ together with the definition of $\beta_{t,\delta}$, *i.e.*, Equation (25), it follows that $T_0(\delta)$ can be upper-bounded by:[21]

$$
\inf\left\{t \in \mathbb{N} : A_0(\delta) + A_1(\eta_1)t^{0.5+\eta_1} + A_2(\eta_2)t^{0.75+\eta_2} \leq t\right\},
$$

where, for brevity, we have shortened $T^\star(\boldsymbol{\mu})\log(1/\delta) + T^\star(\boldsymbol{\mu})K\log(\log(1/\delta) + 1)$ with $A_0(\delta)$. Note that since $\eta_1 < 1/2$ and $\eta_2 < 1/4$, the upper bound is still well-defined and finite.

Next, by applying Young's inequality (Lemma 6), we can further upper-bound this expression as follows:[22]

$$
\inf\left\{t \in \mathbb{N} : A_0(\delta) + \alpha t + \tilde{A}_1(\eta_1, \alpha) + \gamma t + \tilde{A}_2(\eta_2, \gamma) \leq t\right\},
$$

Solving for $t$ yields the desired result. $\qquad \square$

---

[21]Here, for simplicity, we incorporated the $\sqrt{t} + 1$ term and the $K \log\log(t)$ component of $\beta_{t,\delta}$ within the $\sqrt{t \log(t)}$ term.

[22]For the $A_1$ term, apply Lemma 6 with $a = t^{0.5+\eta}, b = A_1(\eta), p = 0.5+\eta, q = 1-0.5-\eta, \epsilon = \alpha/(0.5+\eta)$. Similarly, for the $A_2$ term, use $a = t^{0.75+\eta}, b = A_2(\eta), p = 0.75+\eta, q = 1-0.75-\eta, \epsilon = \gamma/(0.75+\eta)$.

### G.2 AN EXPLICIT BOUND FOR S-TAS

Following the same reasoning that we presented above, it is possible to derive the following explicit expression of $T_0(\delta)$ for S-TAS.

**Proposition 4.** *Consider any $\eta_1 \in (0, 1/2)$, $\eta_2 \in (0, 1/4)$ and let $A_1(\eta_1)$ and $A_2(\eta_2)$ be defined as follows:*

$$A_1(\eta_1) := \frac{80\sigma DLK^2 \log(K)\sqrt{t\log^2(t)}}{\eta_1} \qquad A_2(\eta_2) := \frac{32\sigma D\sqrt{K}T^\star(\boldsymbol{\mu})}{\eta_2}.$$

*Furthermore, consider any $\alpha, \gamma \in (0, 1)$ such that $\alpha + \gamma < 1$ and let $\tilde{A}_1(\eta, \alpha)$ and $\tilde{A}_2(\eta, \gamma)$ be defined as follows:*

$$\tilde{A}_1(\eta_1, \alpha) := A_1(\eta_1)\left(\frac{(0.5 + \eta_1)A_1(\eta_1)}{\alpha}\right)^{\frac{0.5 + \eta_1}{0.5 - \eta_1}} \qquad \tilde{A}_2(\eta_2, \gamma) := A_2(\eta_2)\left(\frac{(0.75 + \eta_2)A_2(\eta_2)}{\gamma}\right)^{\frac{0.75 + \eta_2}{0.25 - \eta_2}}$$

*Then, it holds that:*

$$T_0(\delta) \leq \frac{T^\star(\boldsymbol{\mu})T_{\boldsymbol{\mu}} + T^\star(\boldsymbol{\mu})\log(1/\delta) + T^\star(\boldsymbol{\mu})K\log(\log(1/\delta) + 1) + \tilde{A}_1(\eta_1, \alpha) + \tilde{A}_2(\eta_2, \gamma)}{1 - \alpha - \gamma}.$$

$$(17)$$

*Proof.* The proof is identical to the one of Proposition 3. The main difference is only in the presence of $T^\star(\boldsymbol{\mu})T_{\boldsymbol{\mu}}$ that is due to the additional complexity that affects S-TAS. □

Proposition 4 provides the explicit expression of $T_0(\delta)$ compared to its implicit version that is presented in Theorem 2. Here, comments that are analogous to those that we presented for Proposition 3 hold.

## H AUXILIARY TOOLS

### H.1 TECHNICAL TOOLS

**Lemma 6** (Young's Inequality). *Let $a \geq 0$, $b \geq 0$ and consider integers $p, q > 1$ such that $\frac{1}{p} + \frac{1}{q} = 1$. Furthermore, let $\epsilon > 0$. Then, it holds that:*

$$ab \leq \frac{\epsilon a^p}{p} + \frac{b^q}{q\epsilon^{q/p}}.$$

### H.2 CUMULATIVE TRACKING

The following lemma summarizes the main properties of the C-Tracking procedure.

**Lemma 7** (C-Tracking). *Let $\{\boldsymbol{\omega}(t)\}$ be an arbitrary sequence of elements that belongs to a $K$-dimensional simplex. Consider the C-Tracking applied on a sequence $\{\boldsymbol{\omega}(t)\}_t$ and let us denote by $\tilde{\boldsymbol{\omega}}(t)$ the $l_\infty$ projection of $\boldsymbol{\omega}(t)$ onto $\Delta_K^{\epsilon_t} = \Delta_K \cap [\epsilon_t, 1]$ and $\epsilon_t = (K^2 + t)^{-1/2}/2$. For all $t \in \mathbb{N}$, it holds that:*

$$N_k(t) \geq \sqrt{t + K^2} - 2K \tag{18}$$

$$-K\ln(K)\sqrt{t + K^2} \leq N_k(t) - \sum_{s=1}^{t}\omega_k(s) \leq K\sqrt{t + K^2} \tag{19}$$

$$\sum_{s=1}^{t}\sum_{k\in[K]}\frac{\omega_k(s)}{\sqrt{N_k(t)}} \leq K\ln(K) + 4\sqrt{Kt} + K^2\sqrt{t + K^2} \tag{20}$$

*Proof.* Equation (18) is due to Lemma 7 in Garivier & Kaufmann (2016) and it is due to the forced exploration of the C-Tracking procedure (*i.e.*, the projection onto $\Delta_K^{\epsilon_t}$).

Now, from Theorem 6 in Degenne et al. (2020), we have that:

$$-\ln(K) \le N_k(t) - \sum_{s=1}^{t} \tilde{\omega}_k(s) \le 1. \tag{21}$$

However, from Lemma 7 in Garivier & Kaufmann (2016), we also have that:

$$\max_{k \in [K]} \left| \sum_{s=1}^{t} \omega_k(s) - \tilde{\omega}_k(s) \right| \le K\sqrt{t + K^2}. \tag{22}$$

Combining this result with Equation (21) leads to Equation (19).

Finally, from Lemma 6 in Degenne et al. (2020), we have that:

$$\sum_{s=1}^{t} \sum_{k \in [K]} \frac{\tilde{\omega}_k(s)}{\sqrt{N_k(t)}} \le K\ln(K) + 4\sqrt{Kt}$$

Combining these results with Equation (22) leads to Equation (20), thus concluding the proof. □

## H.3 Properties of Canonical Exponential Families

The following lemma reports standard properties of one-dimensional canonical exponential families.

**Lemma 8** (KL Difference in Exponentialy Families). *For three distributions in a canonical exponential family with means $a, b, c$, it holds that:*

$$d(a, b) = d(a, c) + d(c, b) + (\nu_b - \nu_c)(c - a) \tag{23}$$
$$d(c, b) - d(a, b) \le (\nu_c - \nu_b)(c - a) \tag{24}$$

*where $\nu_{(\cdot)}$ denotes the natural parameter of the distribution with mean $(\cdot)$.*

*Proof.* For a proof, see, *e.g.*, Lemma E.6 in Poiani et al. (2024). □

## H.4 Concentration Results

**Lemma 9** (Good Event). *Consider $\{\mathcal{E}_t\}_t$ such that*

$$\mathcal{E}_t = \left\{ \forall s \in \left[\lceil\sqrt{t}\rceil, t\right], \sum_{k \in [K]} N_k(s)d(\hat{\mu}_k(s), \mu_k) \le 8K\log(s) \right\}$$

*It holds that $\sum_{t=3}^{+\infty} \mathbb{P}_{\boldsymbol{\mu}}(\mathcal{E}_t^c) \le 2eK$.*

*Proof.* The statement is a direct corollary of standard concentration arguments (see Lemma 6 in Degenne et al. (2019)). □

## H.5 $\delta$-Correctness

For simplicity of exposition, we consider the following choice of the threshold $\beta_{t,\delta}$:

$$\beta_{t,\delta} = \log\left(\frac{1}{\delta}\right) + K\log\left(4\log\left(\frac{1}{\delta}\right) + 1\right) + 6K\log(\log(t) + 3). \tag{25}$$

This threshold has been shown to yield $\delta$-correct algorithms for Gaussian distributions Ménard (2019). At a cost of more involved expression, one can adopt the threshold proposed in Kaufmann & Koolen (2021) to analyze the stopping time in generic canonical exponential families.

We now prove for completeness that this choice of $\beta_{t,\delta}$ combined with the stopping and recommendation rules leads to a $\delta$-correct algorithm for any sampling rule. Note that the above result holds for both the cases when $i^\star(\boldsymbol{\mu})$ is both single and multiple-valued.

**Lemma 10** (Correctness). *For any sampling rule and $\boldsymbol{\mu} \in \mathcal{M}$, it holds that $\mathbb{P}_{\boldsymbol{\mu}}(\hat{\imath}_{\tau_\delta} \notin i^\star(\boldsymbol{\mu})) \le \delta$.*

*Proof.* With probabilistic arguments, we have that:

$$\mathbb{P}_{\boldsymbol{\mu}}(\hat{i}_{\tau_\delta} \notin i^\star(\boldsymbol{\mu})) \leq \mathbb{P}_{\boldsymbol{\mu}}\left(\exists t \in \mathbb{N}, j \notin i^\star(\boldsymbol{\mu}) : \inf_{\boldsymbol{\lambda} \in \neg j} \sum_{k \in [K]} N_k(t) d(\hat{\mu}_k(t), \lambda_k) \geq \beta_{t,\delta}\right)$$

$$\leq \mathbb{P}_{\boldsymbol{\mu}}\left(\exists t \in \mathbb{N} : \sum_{k \in [K]} N_k(t) d(\hat{\mu}_k(t), \mu_k) \geq \beta_{t,\delta}\right)$$

$$\leq \delta,$$

where the first step follows from the definition of the stopping and recommendation rules, the second one from the fact that $\boldsymbol{\mu} \in \neg j$ for all $j \notin i^\star(\boldsymbol{\mu})$, and the third one from Proposition 1 in Ménard (2019). $\qquad\square$

