# OpenReview forum: "Non-Asymptotic Analysis of (Sticky) Track-and-Stop"
_ICLR.cc/2026/Conference — ICLR 2026 Oral_

### Official Review · Reviewer_pnVY · 2025-10-31

**Soundness:** 3
**Presentation:** 2
**Contribution:** 3
**Rating:** 6
**Confidence:** 4

**Summary:**

The paper presents a non-asymptotic analysis of the performance of the track-and-stop and sticky track-and-stop algorithms for pure exploration tasks under fixed confidence. The results are non-asymptotic, meaning they hold for confidence levels that do not necessarily converge to zero. The main results are given in Theorems 1 and 2, where the sample complexity upper bounds are stated in an implicit form. More explicit expressions are provided in Appendix G.

**Strengths:**

•	The problem of pure exploration in moderate-confidence regimes is notoriously challenging and worth investigating—see “The Simulator: Understanding Adaptive Sampling in the Moderate-Confidence Regime” by Simchowitz et al., COLT 2017, for relevant discussions (this paper should likely be cited and discussed). Deriving guarantees in this regime is non-trivial, and this appears to be what the authors have successfully achieved here.
•	The upper bounds, although implicit, seem to be more interpretable and user-friendly than those derived in existing works such as Degenne et al. (2019).

**Weaknesses:**

•	There is no comparison with existing results in moderate-confidence regimes (e.g., Degenne et al., 2019). This is an important drawback—the paper focuses on the track-and-stop algorithm, but it remains important to compare its performance with other approaches.
•	There is no numerical experiments! How should we know whether the new upper bounds are reasonable?

**Questions:**

•	I find it hard to believe that no assumption on the mapping $i^\star$ is required. Is the analysis valid for any function, even without regularity assumptions? Does the characteristic time $T^\star(\mu)$ become infinite when regularity is absent?

---

> ### Author Response · Authors · 2025-11-19
> **Rebuttal**
>
> We thank the reviewer for the effort and time taken to review our paper.  Below, we address the remaining concerns and questions.
>
> > The problem of pure exploration in moderate-confidence regimes is notoriously challenging and worth investigating, see, "The Simulator: Understanding Adaptive Sampling in the Moderate-Confidence Regime" by Simchowitz et al., COLT 2017, for relevant discussions (this paper should likely be cited and discussed).
>
> We thank the reviewer for spotting the missing reference, which we will include in our revision. For completeness, we report a discussion below.
>
> In "Simchowitz et al.," 2017, the authors highlighted challenges related to obtaining a tight dependency not only for $\log(1/\delta)$, but also in other parameters (e.g., in $K$).  In our paper, instead, we show that both TaS and S-TaS obtain finite-confidence guarantees that are optimal up to poly-log factors. This is in the same spirit of, e.g., "Non-asymptotic pure exploration by solving games", Degenne et al., 2019 [1].
>
> > There is no comparison with existing results in moderate-confidence regimes (e.g., Degenne et al., 2019). This is an important drawback—the paper focuses on the track-and-stop algorithm, but it remains important to compare its performance with other approaches
>
> We thank the reviewer for raising this point, and we will expand the comparison in the revised version of the manuscript. For completeness, we report the theoretical guarantees below. In brevity, none of the analyses dominate the other.
>
> Both in our results and [1], the main component of the sample complexity is $T_0(\delta)$ that can be expressed as the minimum time $t$ for which $\beta_{t,\delta} \le t T^{*}(\mu)^{-1} - g(t)$. The function $g(t)$ is thus what differentiates the results. Specifically:
> - In our case, $g(t)$ grows, in the worst case, with $t^{3/4}$, while the growth rate of Degenne et al., 2019 is of the order $t^{1/2}$ (hence better).
> - Their $g(t)$ depends linearly on the answer space (that can be exponential in $K$, e.g., in Thresholding bandits). Contrary, we do not have dependency on the answer space.
> - Their $g(t)$ also depends on $\frac{1}{\epsilon}$ where $\epsilon$ is a problem dependent quantity that is $\inf_{\lambda \in \neg i^{\star}(\mu)} \max_k \frac{(\mu_k - \lambda_k)^2}{2\sigma^2}$. This can be significantly small in some problems (e.g., in a BAI task with two arms is roughly $\Delta^2$ where $\Delta$ is the gap between the two arms). Our results do not suffer from this dependence on $1/\epsilon$.
> - The remaining terms (i.e., $K$, $L$, $D$, and $\sigma$), the dependency in both functions is the same up to log factors.
>
> Finally, note that the comparison only holds for single-answer problems. For problems with multiple correct answers, the methods of [1] cannot be applied, and ours are the first results in this general setting.
>
> > There is no numerical experiments! How should we know whether the new upper bounds are reasonable?
>
> While evaluating the asymptotic bound for $\delta \to 0$ is reasonable since the results are easily interpretable (e.g., Table 1 in [2]), it is uncommon to numerically evaluate the finite confidence bound (for instance, neither [1] nor [2] report the finite confidence bound in the experiments). This is due to the fact that, e.g, constants and polylog factors might numerically prevail for large $\delta$. However, due to the tightness of our results, we are guaranteed the same behavior observed in Table 1 of [2], i.e., the convergence of the sample complexity to the fundamental limit of the lower bound.
>
> > I find it hard to believe that no assumption on the mapping $i^{\ast}$ is required. Is the analysis valid for any function, even without regularity assumptions? Does the characteristic time become infinite when regularity is absent?
>
> The reviewer is right; there is, indeed, a regularity condition imposed on $i^{\ast}$. The condition (Lines 131-132) asks that for each $\mu$, there exists $i \in i^{\ast}(\mu)$ such that $\mu \notin \textup{cl}(\neg i)$. Intuitively, this requires that for at least one correct answer $i$, $\mu$ is separated from the set of bandits for which $i$ is not a correct answer, i.e., $\textup{cl}(\neg i)$. This is sufficient to ensure that the lower bound evaluates to a strictly positive quantity (hence learning is possible). Note that discontinuity of $i^{\ast}$ is not problematic in itself (as long as it satisfies this property that we mentioned) because $i\_F(\mu)$ remains upper hemicontinous even for $i^{\ast}$ which are not continuous. This was also observed in [3].
>
> [1] Degenne et al., 2019, "Non-Asymptotic Pure Exploration by Solving Games"
>
> [2] Barrier et al., 2022, "A Non-asymptotic Approach to Best-Arm Identification for Gaussian Bandits"
>
> [3] Degenne et al., 2019 "Pure Exploration with Multiple Correct Answers"

---

### Official Review · Reviewer_mo2G · 2025-10-31

**Soundness:** 3
**Presentation:** 3
**Contribution:** 3
**Rating:** 8
**Confidence:** 3

**Summary:**

The paper revisits the popular pure-exploration algorithms, track-and-stop and sticky track-and-stop. These algorithms are known to have optimal performance in the asymptotic regime, i.e., when the error probability $\delta \to 0$. In this work, the authors analyze these classical algorithm in the finite time-regime and provide non-asymptotic bounds on expected stopping time of these algorithms as a function of $\delta$ and $T^{\star}(\mu)$, the problem specific hardness.

**Strengths:**

The paper provides non-asymptotic bounds for these popular pure exploration algorithms, thereby addressing a gap in our understanding of the finite-time performance of these algorithms.

The analysis seems rigorous and is clearly explained. Overall, the paper is good and I like it.

**Weaknesses:**

I don't see any obvjous weakness in the paper, but I have questions as outlined in the next section.

**Questions:**

From the expression of $T_0(\delta)$, it seems that $T^{\star}(\mu) \log(1/\delta)$ becomes the dominant term only for very small values of $\delta = \mathcal{O}(e^{-K})$. For values larger than this, the term $KT^{\star}(\mu) \log(\log(1/\delta))$ is dominant. This implies that the optimality still holds for only sufficiently small $\delta$. I agree that it common in non-asymptotic analyses to have a threshold only beyond which the results. However, $\delta = \mathcal{O}(e^{-K})$ this choice seems really small for any practical purpose. Ofcourse, this also not because of any leading constants as I have just denoted the order. Can the authors elaborate more on this?

How large would you typically expect $T_{\mu}$ to be, or equivalently how small would $\epsilon_{\mu}$ be? Is there some structure on $\mathcal{M}$ that would allow us to have more explicit bounds on $\epsilon_{\mu}$? This might help shed light on the sub-optimality of the current bounds.

---

> ### Author Response · Authors · 2025-11-19
> **Rebuttal**
>
> We thank the reviewer for the effort in reviewing this work. Below, we answer the two questions.
>
> > Can the authors elaborate more on [$T_0(\delta)$]?
>
> We agree with the reviewer that $T_0(\delta)$ approaches the lower bound with a rate that requires $\delta$ to be small w.r.t. $K$. Nonetheless, we note that this is relevant only to claim the exact (asymptotic) optimality of the algorithm. Indeed, our upper bound on $T_0(\delta)$ still shows that TaS and S-TaS match (up to poly-log terms) the fundamental limit. Understanding and charactering whether these poly-log terms are necessary or not (or whether they are a limitation of the algorithm) is definitely an interesting avenue for future research.
>
> > How large would you typically expect $T^{\ast}$ to be, or equivalently how small $\epsilon\_\mu$ would be? Is there some structure on $\mathcal{M}$ that would allow us to have more explicit bounds on $\epsilon_\mu$?
>
> $T^{\ast}(\mu)$ can, in principle, be arbitrarily large since it depends on the correct answer correspondence and the problem $\mu$ at hand. It can explode even in the $\epsilon$-best arm identification problem on an instance with two equally optimal arms; here, as $\epsilon \to 0$, we have $T^{\ast}(\mu) \to \infty$. But this is a fundamental limit that any algorithm cannot overcome.
>
> Regarding $\epsilon\_\mu$, it has the following interpretation: it measures "how much" upper hemicontinuous the correspondence $i\_F(\mu)$ is at $\mu$. Moreover, in specific instances, one can further characterize this quantity. For example, for the relevant case of the $\epsilon$-best arm identification problem, where $i\_F(\mu) = \textup{argmax}\_{i \in [K]} \mu\_k$ (see, e.g., [1,2]). Let $\Delta_{\mu} = \min\_{i \notin i\_F(\mu)} \mu\_{\ast} - \mu_i$ where $\mu_{\ast}$ is the value of any optimal arm. Then, whenever $\epsilon_\mu < \Delta_\mu$, we have that $i_F(\mu') \subseteq i_F(\mu)$. Hence, one can take, e.g., $\epsilon_\mu = \Delta_\mu/2$.
>
> We thank the reviewer for raising this point, which we find very insightful.
>
> [1] "Non-Asymptotic Sequential Tests for Overlapping Hypotheses Applied to Near Optimal Arm Identification in Bandit Models", Garivier and Kaufmann 2021
>
> [2] "An $\epsilon$-Best-Arm Identification Algorithm for Fixed-Confidence and Beyond", Jourdan et al., 2023

---

### Official Review · Reviewer_Lf7R · 2025-10-31

**Soundness:** 3
**Presentation:** 4
**Contribution:** 1
**Rating:** 4
**Confidence:** 2

**Summary:**

This paper studies the non-asymptotic performance of the Track-and-Stop (TaS) and Sticky-Track-and-Stop (S-TaS) algorithms for best-arm identification. The authors derive upper bounds on the expected number of arm pulls required for both algorithms, using a carefully constructed sequence of events leading to stopping. The main contribution lies in providing finite-time performance guarantees for these two algorithms, complementing their known asymptotic optimality results.

The paper should be rejected due to two main issues:
(1) insufficient contextualization with respect to closely related work, and
(2) a lack of motivation for analyzing the vanilla versions of the algorithms rather than their improved variants.

**Main argumentation**
While the paper is technically sound and written with care, it does not clearly articulate what new insight is gained from analyzing the vanilla (S-)TaS algorithms, given that closely related works already provide non-asymptotic analyzes for more stable variants. Without a stronger justification of novelty or comparison with those results, the contribution remains poor.

**Strengths:**

- The paper is well written and clearly structured.
- The assumptions are well motivated, and the mathematical arguments are rigorous and accompanied by clear intuition.
- The work contributes to a deeper understanding of the stopping behavior of TaS-type algorithms in finite-sample regimes.

**Weaknesses:**

- *Relation to Degenne et al. (2019) and Barrier et al. (2022) is underdeveloped*

Both works introduce algorithmic variants of Track-and-Stop that address its early-phase instability and already provide non-asymptotic analyses.
The paper does not sufficiently motivate why analyzing the vanilla TaS (and S-TaS) provides new insight beyond those results.

The authors claim that the work of Degenne et al. (2019) requires to solve a “more challenging optimization problem,” but do not specify what distinguishes their setting or why this difference matters here.

The discussion of Barrier et al. (2022) is minimal. The paper should clarify in what sense its bounds differ from or improve upon theirs.

- The results are said to complement the observed empirical good performance of TaS, yet no empirical evaluation is provided to assess the tightness or practical relevance of the derived bounds.

- The introduction of S-TaS as a solution to "general multiple-answer problems" is insufficiently motivated. It is unclear what new problem instances this algorithm targets or why these instances are practically or theoretically important.

**Questions:**

- Why is it valuable to analyze the vanilla Track-and-Stop, given that stabilized variants (Degenne et al., Barrier et al.) already offer non-asymptotic guarantees and often perform better empirically?
- What are the “general multiple-answer problems” mentioned, and why can’t previous approaches be extended to handle them?
- In what scenarios would one prefer using TaS or S-TaS over the existing improved variants?
- Have you evaluated whether your non-asymptotic bounds align with the empirical stopping times observed in practice?

---

> ### Author Response · Authors · 2025-11-19
> **Rebuttal**
>
> We thank the reviewer for the effort and time taken to review our work. Before answering the questions, we would like to make a general remark.
>
> The main objection of the reviewer is the lack of justification for our contributions and, in particular, the reason why we analyze the "vanilla" variants of TaS and S-TaS. We defer a detailed discussion on related works in the next paragraph while in the following we address the concern on the significance of the contributions by remarking to the reviewr that; 1) TaS is one of the most studied and successful algorithms for pure exploration problems with a single correct answer; it is also the algorithm that since the work of Garivier et al., 2016, jumpstarted the line of works on asymptotically optimal pure exploration methods and 2) S-TaS is the only algorithm, to the best of our knoledge, that can handle generic problems with multiple correct answers, and thus our analysis is the first providing finite confidence guarantees for these general problems (note that neither TaS nor its variants of [2] and [3] can deal with this setting).
> In light of these points, we believe that it is fair to say that our theoretical contributions fill an important gap in our understanding of these important algorithms.
>
>
> > Why is it valuable to analyze the vanilla Track-and-Stop, given that stabilized variants (Degenne et al., Barrier et al.) already offer non-asymptotic guarantees and often perform better empirically? [Also the] Relation to Degenne et al. (2019) and Barrier et al. (2022) is underdeveloped
>
> It is valuable to analyze the vanilla algorithm as the two variants are considerably more complex (introducing additional computational overhead in each iteration, as detailed below) and do not consistently outperform TaS (see discussion below). Our results, in particular, demonstrate that the proposed modifications are unnecessary if one is concerned with finite-confidence performance. This finding is striking, as both [2] and [3] introduce substantial algorithmic complications to obtain these guarantees.
>
> - The optimistic TaS variant of Degenne et al., 2019 [2] is significantly more computationally demanding than TaS. At each round, it requires evaluating optimistically the lower bound across all models in a confidence region around the estimated $\hat\mu(t)$, leading to a *max-max-min* problem which, in general, does not have efficient oracles. We emphasize that, in contrast, TaS only requires maximizing a concave function over the simplex, which is much more computationally manageable.
>
> - The stabilized version of Barrier et al., 2022 [3] usually underperforms TaS. As shown in that paper, TaS achieves better performance across all instances on which they were tested. Furthermore, their algorithm is restricted to Gaussian and unstructured best-arm identification problems. Here, we deal with arbitrary structured single-answer problems (i.e., in our analysis of TaS, the task can be any single-answer correspondence and $\mathcal{M}$ can encode arbitrary structures).
>
> We thank the reviewer for raising this point. In case of acceptance, we will make use of the additional space to expand the comparison with [2] and [3].
>
>
> > What are the “general multiple-answer problems” mentioned, and why can’t previous approaches be extended to handle them?
>
> We thank the reviewer for raising this clarity issue. We will modify the revised version to clarify the meaning of general multiple-answer problems.
>
> In particular, the meaning of "general multiple-answer problems" is two-fold:
> - **Arbitary structure**: We can deal with arbitrary bandit structures that can be encoded through $\mathcal{M}$, which is a known subset of all possible bandit models. For example, $\mathcal{M}$ could be the set of Lipschitz bandits with finite arms, dueling bandits, or unimodal bandits.
> - **Arbitrary multiple correct-answer correspondences**: S-TaS is able to take in input any task (i.e., any multiple correct-answer correspondence) and yield an optimal sampling procedure. Example of tasks that are encompassed by this framework are thresholding bandits, $\epsilon$-minimum threshold, any low arm, $\epsilon$-best arm (see also [1] for further examples).
>
> The reason why methods for single-answer problems (e.g., TaS and also [2] and [3]) cannot be adopted in these settings has been discussed thoroughly in [1]. The problem is related to the presence of multiple values in the correct-answer correspondence $i^*$. As we discuss in the background section, the main reasons are the upper hemicontinuity of the $i_F(\mu)$ correspondence, and the non-convexity of the set $\omega^{\ast}(\mu)$. Both of these complications cannot be handled by standard single-answer approaches such as TaS or its variants presented in [2] and [3]. To the best of our knowledge, S-TaS is the only known effective algorithm for this general class of problems.

---

> ### Author Response · Authors · 2025-11-19
> **Rebuttal (cont'd)**
>
> > In what scenarios would one prefer using TaS or S-TaS over the existing improved variants?
>
> In the interest of brevity, we only summarize the distinctions with prior works, as we already discussed the differences in the previous answers.
>
> **Multiple-answer problems: S-TaS is the only option**
>
> As we discussed above, when the problem admits multiple correct answers, Sticky-TaS (S-TaS) remains the only optimal algorithm for general pure exploration settings. In this work, we provide the first finite-confidence guarantees for S-TaS.
>
>
> **Single-answer problems: TaS vs. its variants**
>
> When the problem admits a single correct answer, TaS should be preferred over Optimistic-TaS [2] whenever implementing the max–max–min oracle is too computationally demanding.
>
> Regarding the variant proposed by Barrier et al., as acknowledged by the authors, TaS typically achieves better empirical performance. Moreover, our results show that TaS preserves finite-confidence guarantees on a broader class of problems with respect to Barrier et al., since our framework accommodates arbitrary structures.
>
> > Have you evaluated whether your non-asymptotic bounds align with the empirical stopping times observed in practice?
>
>
> While evaluating the asymptotic bound for $\delta \to 0$ is reasonable since the results are easily interpretable (e.g., Table 1 in [3]), it is uncommon to numerically evaluate the finite confidence bound (for instance, neither [3] nor [2] report the finite confidence bounds in the experiments). This is due to the fact that, e.g, constants and polylog factors, might numerically prevail for large $\delta$. However, due to the tightness of our results, we are guaranteed the same behavior observed in Table 1 of [3], i.e., the convergence of the sample complexity to the fundamental limit of the lower bound.
>
>
> [1] Degenne et al., 2019 "Pure Exploretion with Multiple Correct Answer"
>
> [2] Degenne et al., 2019, "Non-Asymptotic Pure Exploration by Solving Games"
>
> [3] Barrier et al., 2022, "A Non-asymptotic Approach to Best-Arm Identification for Gaussian Bandits"

---

> > ### Comment · Reviewer_Lf7R · 2025-11-20
> >
> > Thank you for the thorough rebuttal. My two main concerns have been addressed well, and I will update my score. I encourage the authors to reflect this discussion in the camera-ready.

---

### Official Review · Reviewer_41vV · 2025-11-02

**Soundness:** 3
**Presentation:** 3
**Contribution:** 3
**Rating:** 6
**Confidence:** 3

**Summary:**

This paper studies the non-asymptotic behavior of the Track and Stop algorithm, along with its variant, Sticky Track and Stop, which handles cases with multiple correct answers. This is a heavily theoretical paper. While Track and Stop has been around for several years and its empirical performance has been repeatedly validated showing it to be a strong and hard-to-beat baseline, its non-asymptotic analysis has been known to be challenging. This paper serves as a strong addition to the field, with its primary contribution being the non-asymptotic analysis for both the standard algorithm and its 'Sticky' variant.

**Strengths:**

1. The paper is well-written, with a clear structure, strong motivation, and a thorough discussion of related work. Overall, it is very easy to follow.
2. The theoretical result is a welcome addition to this field, as a non-asymptotic analysis for Track and Stop has been long-missing.

**Weaknesses:**

I did not find any obvious weaknesses. However, due to time constraints, I was unable to review the proof details thoroughly.

**Questions:**

For the 'Sticky' variant, which addresses multiple correct answers, a comparison to the findings in "Revisiting simple regret: Fast rates for returning a good arm" (ICML 2023) would be insightful. That work, though in a fixed-budget setting, discovered that the identification speed can be significantly faster (potentially at a $1/M$ rate, where $M$ is the number of good arms). It would be interesting for the authors to discuss whether a similar, weaker, or analogous result is possible in the fixed-confidence setting. This is currently difficult to determine, as the complexity bounds for Sticky Track and Stop in this paper are not expressed in an explicit form that allows for an easy comparison.

---

> ### Author Response · Authors · 2025-11-19
> **Rebuttal**
>
> We thank the reviewer for the time taken to review our paper. Below, we answer the remaining question.
>
> > For the 'Sticky' variant, which addresses multiple correct answers, a comparison to the findings in "Revisiting simple regret: Fast rates for returning a good arm" (ICML 2023) would be insightful.
>
> We thank the reviewer for the reference. Even though a similar advantage could be expected in our setting, this is not the case. Indeed, when focusing on the identification of an $\epsilon$-optimal arm, the behavior of the complexity as a function of the number of $\epsilon$-optimal arms is captured in our bounds by $T^* (\mu)$. In particular, from the results of [1] we know that $T^* (\mu) \in \Theta( \sum\_\{i \ne i^* \} \frac{1}{(\mu_* - \mu_i + \epsilon)^2} )$ where $i^*$ is any optimal arm $\mu\_{\ast}$. Thus, this improvement does not yield the same advantage observed in the fixed-budget setting, since this quantity depends on the (adjusted) gap of all arms, and not only on the number of good arms. However, it is interesting to note that this characterization still shows that the problem becomes easier as $\epsilon$ increases.
>
> To conclude, improving our general results in specialized settings like the one suggested by the reviewer represents an interesting avenue for future research but lies beyond the scope of the present paper. We will highlight this as an interesting open problem within the conclusion section.
>
> [1] "Non-Asymptotic Sequential Tests for Overlapping Hypotheses Applied to Near Optimal Arm Identification in Bandit Models", Garivier and Kaufmann, 2021

---

### Author Response · Authors · 2025-12-02

Dear AC,

Given the recent challenging circumstances, we prepared a brief summary of the rebuttal outcome to support your assessment and to highlight the effort we dedicated to addressing the reviewers’ concerns.

In particular, Reviewer Lf7R raised several valuable points in their initial review, especially concerning the connection to prior TaS variants, the motivation for studying the basic TaS and Sticky-TaS algorithms, and the range of problems our framework can handle. In the rebuttal, we addressed each of these points with detailed explanations and clarifications. After considering our responses, Reviewer Lf7R increased their score substantially, from 4 to 8, explicitly stating that their concerns had been satisfactorily resolved. After the discussion, the final score was 8, 8, 6, 6.

We understand that this is an unusually difficult situation for the committee, and we sincerely appreciate the extra care being taken to ensure fairness and consistency in the decision process. We hope that this concise summary is useful, and we thank you for your attention and effort.

---

### Meta-Review · Area_Chair_WRi2 · 2025-12-08

**Summary:**

This can be treated as a unanimous recommendation for acceptance, as the only score below the acceptance threshold had a follow-up comment stating that they will increase it.  The reviewers commented positively on most aspects of the paper, in particularly highlighting how it closes an important gap in the literature on a widely-adopted algorithm with strong theoretical guarantees.

**Reviewer Concerns:**

There are no notable remaining concerns sufficient to change the decision.

**Reviewer Scores:**

The likely final scores would be 8/6/6/6, or if lucky maybe even 8/8/6/6.

---

### Decision · Program_Chairs · 2026-01-26

Accept (Oral)